# Conformal in-ear bioelectronics for visual and auditory brain-computer interfaces

Zhouheng Wang[1,2,8], Nanlin Shi[3,8], Yingchao Zhang[2], Ning Zheng[4], Haicheng Li[1,2], Yang Jiao[1,2], Jiahui Cheng[1,2], Yutong Wang[1,2], Xiaoqing Zhang[5], Ying Chen[6], Yihao Chen[1,2], Heling Wang[1,2], Tao Xie[4], Yijun Wang[7], Yinji Ma[1,2] ✉, Xiaorong Gao[3] ✉ & Xue Feng[1,2] ✉

Brain-computer interfaces (BCIs) have attracted considerable attention in motor and language rehabilitation. Most devices use cap-based non-invasive, headband-based commercial products or microneedle-based invasive approaches, which are constrained for inconvenience, limited applications, inflammation risks and even irreversible damage to soft tissues. Here, we propose in-ear visual and auditory BCIs based on in-ear bioelectronics, named as SpiralE, which can adaptively expand and spiral along the auditory meatus under electrothermal actuation to ensure conformal contact. Participants achieve offline accuracies of 95% in 9-target steady state visual evoked potential (SSVEP) BCI classification and type target phrases successfully in a calibration-free 40-target online SSVEP speller experiment. Interestingly, in-ear SSVEPs exhibit significant 2nd harmonic tendencies, indicating that in-ear sensing may be complementary for studying harmonic spatial distributions in SSVEP studies. Moreover, natural speech auditory classification accuracy can reach 84% in cocktail party experiments. The SpiralE provides innovative concepts for designing 3D flexible bioelectronics and assists the development of biomedical engineering and neural monitoring.

In recent years, BCIs have shown considerable potential in motor and language rehabilitation for patients with clinical needs, leading to the development of innovative approaches for healthy individuals to achieve efficient, natural and intelligent human-machine interactions[1–6]. However, most BCIs involve cap-based non-invasive methods[7,8], headband-based commercial products[9,10], or microneedle-based invasive methods[11–15], which are not acceptable for daily use due to inconvenience and inflammation risks and even irreversible damage to soft tissues. Recently, BCIs related to ears have drawn increasing attention because of its multifold advantages: (i) the inherent advantages of achieving gel-free acquisition in hairless areas; (ii) the non-invasive electroencephalogram (EEG) signal acquisition makes it possible for developing wearable and discreet neural state monitoring electronics used in daily life. This type of EEG detection is known as ear EEG[16,17].

With the development of bioelectronics[18–28], several ear EEG monitoring electronics have been proposed and have achieved remarkable results[29–31]. Among them, in-ear EEG monitoring has attracted more attention stemming from its unique advantages of wearability and discreteness. Several challenges associated with in-ear

[1]Laboratory of Flexible Electronics Technology, Tsinghua University, Beijing 100084, China. [2]AML, Department of Engineering Mechanics, Tsinghua University, Beijing 100084, China. [3]Department of Biomedical Engineering, Tsinghua University, Beijing 100084, China. [4]State Key Laboratory of Chemical Engineering, College of Chemical and Biological Engineering, Zhejiang University, Hangzhou 310027, China. [5]Department of Otolaryngology-Head and Neck Surgery, Beijing Tongren Hospital, Capital Medical University, Beijing 100730, China. [6]Institute of Flexible Electronics Technology of THU, Zhejiang, Jiaxing 314000, China. [7]Institute of Semiconductors, Chinese Academy of Sciences, Beijing 100083, China. [8]These authors contributed equally: Zhouheng Wang, Nanlin Shi. ✉e-mail: mayinji@tsinghua.edu.cn; gxr-dea@mail.tsinghua.edu.cn; fengxue@tsinghua.edu.cn

EEG detection should be resolved for further applications, one of which is the geometrical complexity and physiology sensitivity of the ear canal. The shape of the canal is circuitous and varies from person to person; thus, to form a tight electrodes-tissue interface, electronics must be deformable and adaptive. However, most of the current electronics require supports such as earplugs or 3D printed attachments to ensure that the electrodes are in good contact with the inner wall of ear canal. Although promising, some common problems still exist: (i) interfaces failure and electrodes destruction frequently occur due to integrating curved support surfaces with electrodes based on traditional planar processing technology; (ii) the solid supports hinder subjects from communicating with the outside world; (iii) it is hard to meet the opposite requirements of tight contact and comfort. Specifically, hard, non-deformable supports cannot compatibly deform along the circuitous ear canal, which often result in poor local interfacial adhesion between the electrodes and the inner canal. On the other hand, soft supports which are deformable and conformable are also proposed to replace the hard one, but the strong friction with the sensitive canal causes irritation or even inflammation to ears during the inset and removal processes.

Here, we develop in-ear bioelectronics, named as SpiralE, which can adaptively expand and spiral along the auditory meatus under electrothermal actuation, thus ensuring conformal contact while avoiding over-constrain on the meatus. As a result, EEG can be stably recorded with the proposed discreet and wearable bioelectronics without affecting the subjects' communication with the outside world while reducing friction against the inner wall of ear canal owing to the smaller contact area (50 mm × 3 mm) and lower modulus during the deformation and removal processes. Based on SpiralE, we propose in-ear BCI paradigms that are comprised of SSVEP-BCI, which decodes overt visual attention, and auditory BCI, which decodes covert auditory attention in natural speech scenarios. The results of the visual experiments reveal that while the fundamental frequency is primarily localized in the occipital area, the distribution of harmonics shows a greater inclination towards the temporal-parietal region. These results provide valuable information for explaining the cause and spatial propagation of the harmonics. Moreover, the decoding accuracies of the in-ear channels are 95% in the offline 9-target SSVEP task and 75% in the online 40-target SSVEP task without training. The Information Transfer Rate (ITR) reaches 36.86 ± 15.53 bits/min, which is the highest to those reported in the previous ear EEG results, demonstrating the potential of building an efficient communication pathway based on SpiralE. More interestingly, the auditory classification accuracy in natural speech can reach 84% in cocktail party experiments owing to the hollowness of SpiralE, suggesting that our design could provide reliable auditory attention decoding (AAD) for realizing auditory BCI in natural scenes.

## Results

### Design and overview of SpiralE

Since each person's auditory meatus has a different shape and size and the internal structure is tortuous, we introduce our SpiralE in a compressed temporary shape (a smaller spiral form) into the ear canal (upper-left inset of Fig. 1a). Then we trigger the shape memory effect by applying Joule heating generated by an external electric field, and resulting in SpiralE expansion to a predetermined spiral shape with a larger radius (lower-left inset of Fig. 1a). To make SpiralE form a tight contact with the inner wall of ear canal, the radius of the predetermined spiral is designed to be larger than that of the ear canal. During the shape recovery process, SpiralE will be impeded by the wall of the ear canal; thus, SpiralE can conform to the shape of the auditory meatus, effectively addressing the user-to-user variation and complexity of ear canals. In addition, SpiralE remains self-supported on the inner wall of the canal after the stimulation is removed and cooling down to the body temperature, thus achieving a conformal fit and ensuring the stability and reliability of ear EEG

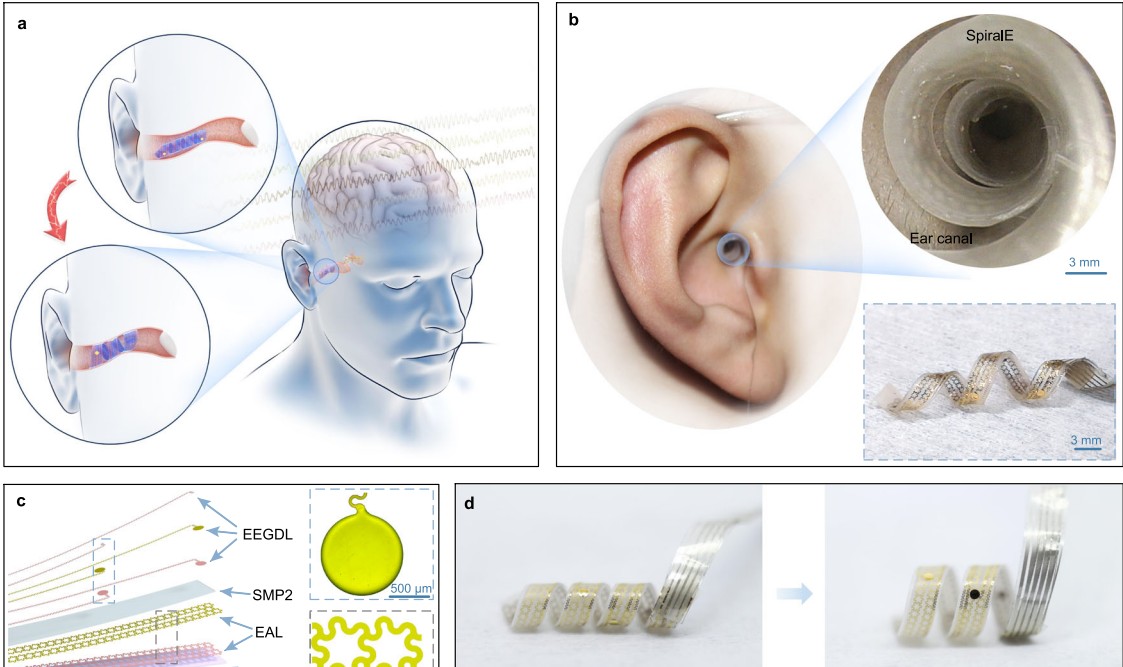

**Fig. 1 | Design of in-ear bioelectronics. a** Schematic diagram of in-ear EEG recording by SpiralE. **b** Pictures of SpiralE conformally adapting to the inner wall of the ear canal. Upper-right inset is a photograph captured by a medical endoscope. Lower-right inset shows the irregular three-dimensional structure of SpiralE after removal form ear. **c** Exploded-view schematic illustration of the functional layers of the proposed SpiralE. Right insets are photographs of EEG detection layer (EEGDL) and electrothermal actuation layer (EAL), respectively. **d** SpiralE in temporarily fixed shape (left) and recovers to permanent shape (right) with a larger radius.

detection (left of Fig. 1b). The shape of SpiralE in ear has been photographed using a medical endoscope (OLYMPUS Visera Elite), as shown in upper-right inset of Fig. 1b, demonstrating that SpiralE can well adapt to the inner wall of ear canal with a hollow form. Furthermore, we carefully pull out SpiralE from ear after in-ear EEG recording, and find that SpiralE shows an irregular three-dimensional structure with a variable radius along the axial direction, which is consistent with the shape of the ear canal (lower-right inset of Fig. 1b). After electrothermal actuation, SpiralE continues to deform until it returns to the permanent large spiral configuration (Supplementary Movie 1). This further indicates that SpiralE does not fully recover to the designed uniform radius in ears after stimulation, but adaptively conform to the inner wall of ear canal.

More importantly, the decrease-increase-decrease in the modulus of SpiralE during the insertion-deformation-detection-extraction processes not only ensures close contact with ear canal during monitoring but also reduces the discomfort caused by friction against the canal wall during the expansion and extraction processes, as shown in Supplementary Fig. 1. On the one hand, the deformation of SpiralE in the ear and the removal from the ear are performed at a lower modulus, thereby mitigating the potential adverse effects on the ear canal. On the other hand, the high modulus provides adequate support to conform to the ear canal, thereby reducing the instability of EEG monitoring caused by motion artifacts. In addition, SpiralE has been designed as a long strip with an in-plane size of $50\,mm \times 3\,mm$ (where the electrode is a circle with a diameter of $1\,mm$) and is supported on the ear canal with a spiral shape to finish in-ear EEG recording, rather than the previous devices that have full contact with the inner wall of the ear canal (Supplementary Table 1). This strategy further reduces abrasion and friction with the ear canal. As a comparison, we summarize existing ear EEG electronics and SpiralE in terms of their sensing units and application scenarios, as shown in Supplementary Table 1, to further demonstrate advantages of SpiralE.

The main components of SpiralE include double-layer shape memory polymers (SMPs) embedded by an electrothermal actuation layer (EAL) and an EEG detection layer (EEGDL), as shown in Fig. 1c. The top layer (namely, the outer layer in 3D spiral state) is EEGDL, which is comprised of conducting Au wires and insulating polyimide (PI). The exposed conducting sites function as signals recording (upper-right inset of Fig. 1c). Next to EEGDL is the SMP2 layer, which has a good surface adhesion in the rubber state, thus assisting in the deformation of SpiralE and the bonding EEGDL and EAL to ensure that the structure does not delaminate and fail during the deformation process[32]. The SMP1 layer with relatively high modulus acts as a deformable skeleton to realize the deformation of the bioelectronics from a small spiral to a large one with high strength. A description of the two kinds of SMPs is presented in Supplementary Figs. 2–3 and the Methods section. The EAL, in the middle of the SMP1 and SMP2, is comprised of a stretchable and flexible conductive mesh (lower-right inset of Fig. 1c) to generate Joule heating under an external electric field. The application of EAL is introduced to generate heating as demand instead of using hot water to drive the deformation as described in previous studies[33], which is not suitable for EEG detection in ears.

The details about the preparation of the materials and functional layers are described in Methods. Briefly speaking, the fabrication paradigm includes the formation of two types of planar flexible and stretchable electrodes via traditional 2D micro/nanotechnology (Supplementary Figs. 4a–c), assembly of four layers by multistep transfer printing (Supplementary Fig. 4d), reconfiguration of permanent shape into a spiral with pre-designed radius of $r_0$ (Supplementary Figs. 4e, f), and program to a temporary shape (Supplementary Figs. 4g, h), i.e., a small spiral shape with a radius of $r_1$ ($r_1 < r_0$). An in vitro experiment of the entire shape recovery and expanding deformation of the developed SpiralE is shown in Fig. 1d.

## Validation and characterization of SpiralE

To validate the fatigue characteristics of SpiralE, we investigate the dynamic resistance modulation of its most vulnerable constituents, namely, the stretchable electrodes (comprising the EAL and EEGDL), subjected to 400 cycles of 10% tensile strain, as depicted in Supplementary Fig. 5. Our analysis ascertains the efficacy of SpiralE, as evidenced by the consistent resistance modulation observed during the stretching process and negligible alteration in resistance post-stretching. These results corroborate the feasibility of utilizing SpiralE in repetitive or short-term applications, such as for a handful measurements.

Electrical impedance of electronics is critical for increasing the recorded signal-to-noise ratio (SNR). The impedance spectroscopy of the electrodes is obtained according to electrochemical impedance spectroscopy (EIS) under four different states, i.e., the initial flat state, the permanently reconfigured spiral state, the temporary spiral state, and the recovered spiral state. The results show that the values of all channels vary little in these four states (Fig. 2a). In addition, we observe that SpiralE (where the electrode is a circle with a diameter of $1\,mm$) exhibits an impedance magnitude of less than 1 kilohms at 50 Hz. This finding is comparable to the results of previous reports[33,34] and suggests that SpiralE is suitable for use in EEG detection. Moreover, the impedance values in the five channels under repeated deformation to reach the final large spiral state are plotted in Fig. 2b, which verifies the electrochemical stability of SpiralE.

The hollowness of SpiralE ensures that all subjects could hear audio stimuli in the outside world in real time, which is quantitatively characterized in vitro. Figure 2c shows the comparison of decibel values of earplugs, earphones, and SpiralE in water drop sound environments. Due to the hollowness of SpiralE, outside sounds are nondestructively transmitted to subjects' ears in real time, even during the tasks, which can be hardly achieved by in-ear electronics based on headphones and earplugs.

To verify the effectiveness of structural reconfiguration process, a SpiralE is firstly temporarily programed into a small spiral state and then places in warm water at $50\,°C$, and the small spiral quickly recovers to the permanent state, namely, the large spiral shape (Supplementary Movie 2). We also carry out in vitro experiments to simulate the real application of SpiralE in ear canal. A SpiralE in a small spiral has been inserted into a plastic tube, whose inner diameter is $6.5\,mm$ (similar to ear canal). After electrothermal actuation, the SpiralE is heated up above the transition temperature of SMPs, resulting in shape recovery of SpiralE to the permanent state, i.e., expanding to a spiral with a larger radius. Owing to the contact constrain from the tube, SpiralE can fully conform to the inner wall. We extract the maximum inner radius during deformation process and plot in Fig. 2d, demonstrating the active deformation ability of SpiralE in enclosed tube clearly. Supplementary Movie 3 shows the expansion process of SpiralE in vitro. With the increasing of temperature by Joule heating, the modulus of SMPs decreases, leading to the softening of SpiralE. With such a design, we aim to diminish the damage to the inner wall of ear canal during deformation process. Then, SpiralE has been placed in a right-angle elbow and a similar recovery process occurs driving by the Joule heating. It can expand and adaptively conform to the inner wall of the pipe, showing the ability of SpiralE to fit complex nonuniform curvatures under electrothermal actuation, as illustrated in Fig. 2e, Supplementary Movies 4 and 5.

Subsequently, we proceed to insert a SpiralE into a polyethylene (PE) tube with an inner diameter of $6.5\,mm$, followed by electrothermal actuation until the device achieves conformity with the inner wall of the tube. This is performed to obtain a quantitative description of the high-temperature duration. Throughout the duration of the experiment, we employ a Fluke Ti400 thermal imaging camera to monitor both the temperature of the entire field and the deformation of SpiralE, while simultaneously recording time. The resultant time-temperature

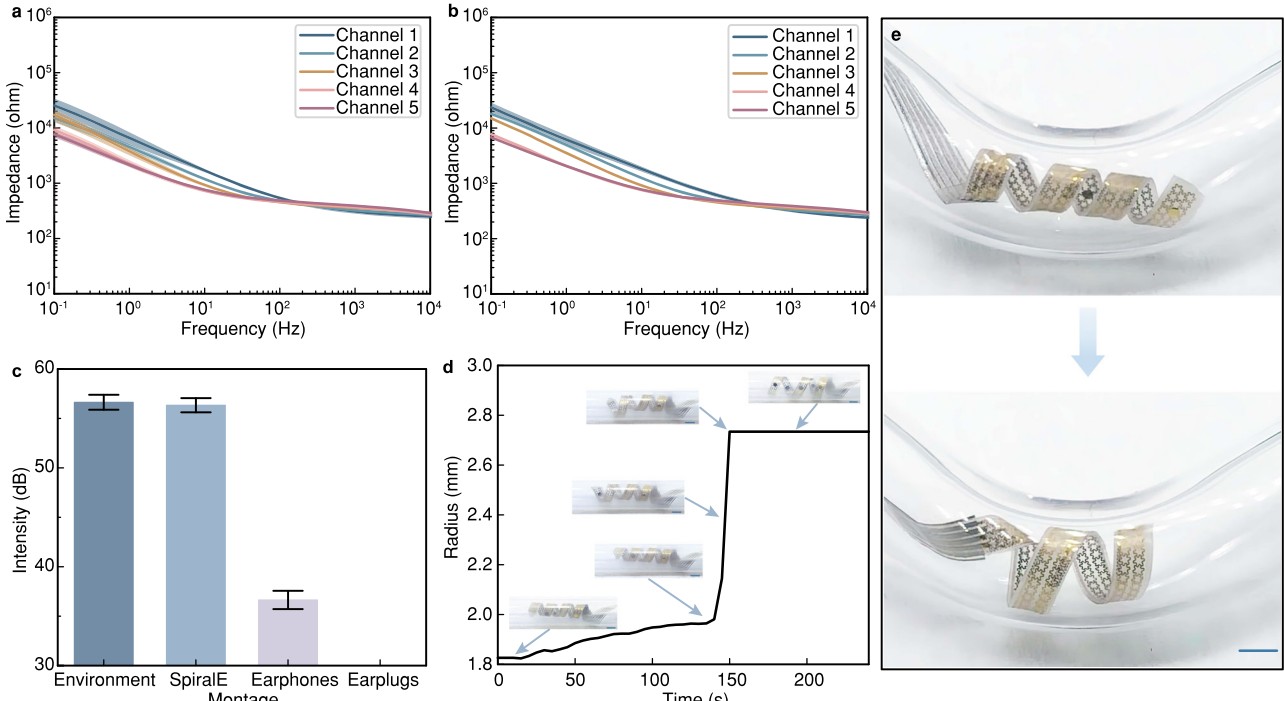

**Fig. 2 | In vitro validation and characterization of SpiralE. a** Electrochemical impedance spectroscopy (EIS) test results of the five channels in four states (*n* = 4, mean, error bands: SD). **b** EIS test results of the five channels during five repeated expansions and deformations (*n* = 5, mean, error bands: SD). **c** The decibel values detected by three types of in-ear electronics in task conditions (*n* = 150, data are presented as mean values +/− SD). **d** The maximum inner radius of SpiralE during deformation process in a tube. The insets show the status in corresponding stages. **e** Images of SpiralE expand and deform at complex curvature radii in a right-angle elbow. Scale bars, 3 mm. Source data from (**a–d**) are provided as a Source data file.

curves are presented in Supplementary Fig. 6, along with corresponding thermal images captured at various stages of the deformation process (indicated by red borders). Notably, the dotted line depicted in the figure indicates that although the temperature of samples 2 and 3 continue to rise, the deformation process has already been completed in the preceding stage. The experiments last between 5 to 10 min, with the maximum temperature in the field exceeding 49 °C for ~30 s, complying with a number of established international standards for allowable maximum temperatures on skin.

It should be noted that the actual temperature perceived by the skin is expected to be significantly lower than the maximum field temperature recorded by the thermal imaging camera, due to the presence of multiple layers including SMP2, EEGDL, and the convective air layer between the EEGDL and skin. In particular, a non-spiral device has been affixed to the forearm, and a sheet-type thermocouple probe is placed directly between the device and the skin. The device is subjected to electrothermal actuation, which uniformly elevates the temperature across the field to 50 °C, ascertained through thermal imaging (Fluke Ti400). Simultaneously, the temperature of the skin surface is measured utilizing a thermocouple (Supplementary Fig. 7a). Upon reaching the peak temperature of 50 °C in the field (Supplementary Fig. 7b), the temperature feedback from the thermocouple affixed to the skin surface is found to be around 40 °C (Supplementary Fig. 7c), thereby confirming the safety of the device during contact with the skin. This data analysis suggests that the device can be utilized without any potential harm to the skin. Moreover, the device has been left attached to the skin for over 3 min at a maximum field temperature of 50 °C, and upon removal, no redness or swelling is observed at the attachment site (Supplementary Fig. 7d).

Additionally, finite element analysis of a SpiralE with an inner radius of 3 mm constrained by an ear canal with a radius of 2.5 mm shows that the interfacial stresses on the skin in contact with SpiralE are within the normal skin perception threshold (20 kPa)[35,36], as

depicted in Supplementary Fig. 8. As a result, SpiralE is unlikely to impose an unacceptable burden on this sensitive region.

Ear EEG has shown feasibility for epilepsy detection, sleep monitoring, and drowsiness detection[37,38]. Such applications have huge potentials to facilitate convenient neural state monitoring for both clinical and real-life scenarios. Here we manifest such potentials of our SpiralE by building an efficient communication pathway of SSVEP speller and an intelligent auditory attention decoder to classify attended natural speech. The relevant setups and procedures for the in vivo experiments are described in detail in the Methods.

In this study, we also employ a Fluke Ti400 thermal imaging camera to examine the maximum temperature of the device subsequent to insertion into the ear. Our results indicate that the maximum field temperature during SpiralE's deformation is 49.3 °C (Supplementary Fig. 10), while the temperature sensed by the skin is within the range of 40 °C, as confirmed by the above experiments. All subjects report feeling normal after SpiralE has been supported on the auditory meatus and successfully complete the experiments, confirming the safety of our device for use in ears.

Supplementary Fig. 11 displays the SpiralE-skin impedance of in-ear channels, with and without gel, as determined via an Impedance Analysis Interface (PSM 1700, Newtons4th Ltd). The impedance at 50 Hz has been determined to be 110.96 ± 23.31 kilohms across all in-ear channels, and the relevant data has been presented in Supplementary Table 1. Despite the convenience and enhanced stability of electrophysiological contact provided by the gel, our experiments demonstrate that the conformed SpiralE without gel still produces robust EEG data. In addition, the impedance of the SpiralE-skin has been recorded during the experimental trials utilizing a commercial EEG recorder ("Methods") to demonstrate the assessment of contact stability and EEG reliability. An illustrative example of such recordings for a participant is presented in Supplementary Fig. 12. The impedance values of all subjects during the alpha rhythm and SSVEP recordings

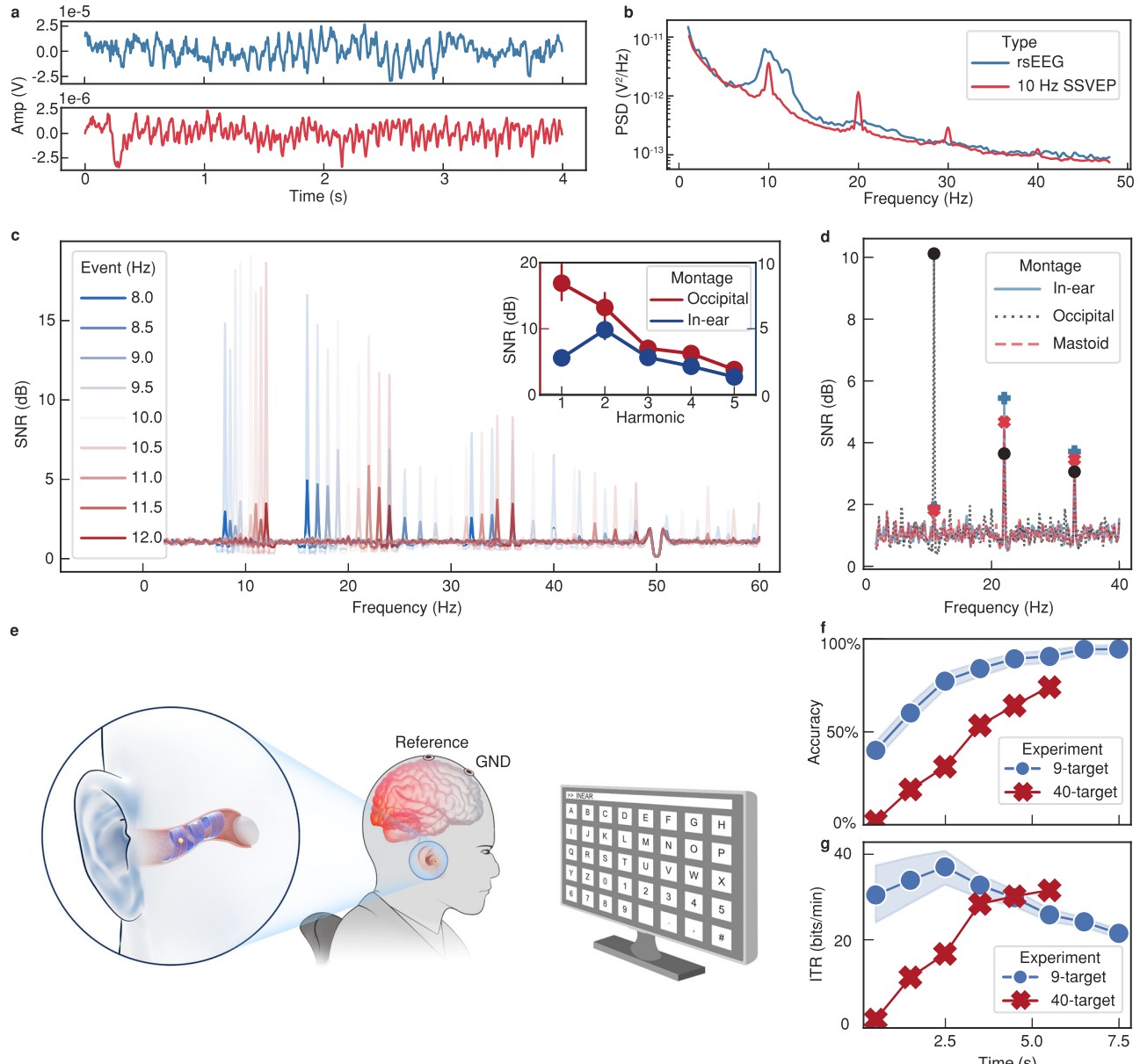

**Fig. 3 | Visual BCIs based on SpiralE. a** Channel averaged time waveforms for in-ear EEG in alpha rhythm (top) and 10 Hz SSVEP recordings (bottom) for one subject. **b** The group-level and channel averaged in-ear EEG power spectral density (PSD) of the alpha rhythm and 10 Hz SSVEP recordings. **c** The 9-target SSVEP SNR of occipital EEG and in-ear EEG in light and dark shades. The illustration in the upper-right corner describes the harmonic SNR of both montages ($n = 9$, mean, error bars: 95% confidence intervals). **d** Spectral comparison of the in-ear EEG, mastoid EEG, and occipital EEG signals of one subject. **e** Schematic diagram of the 40-target online SSVEP-BCI with SpiralE. **f, g** The accuracy and ITR in the 9-target offline SSVEP-BCI and 40-target online SSVEP-BCI experiments ($n = 9$ for 9-target offline SSVEP-BCI, mean, error bands: 95% confidence intervals).

---

are displayed in Supplementary Table 2, and the available electrodes (<50 kilohms) exceed 80% (8/10) for the majorities of recordings.

## Visual BCIs based on SpiralE

To validate SpiralE can act as a possible BCI device, we first record the alpha rhythms as a benchmark for SpiralE evaluation. Alpha rhythms are endogenous oscillations in frequency range of 8–13 Hz that are primarily found in the occipital and parietal lobes in relaxed eye-closed states and are associated with various perceptual effects[39]. As shown by the blue lines in Fig. 3a, b, a portion of the temporal waveform and the subject-averaged spectral analysis confirm the strong oscillatory alpha power in the resting-state EEG (rsEEG). The temporal and spectral dimensions of rsEEG demonstrate that SpiralE can capture clear alpha oscillations centered at ~10 Hz. Individual spectral

representations for all subjects are shown in Supplementary Fig. 13 by the blue lines.

We then build SSVEP-BCIs to demonstrate the advantages of the proposed SpiralE. SSVEPs are periodic neural responses evoked by periodic visual stimuli at specific frequencies[7]. As the response mainly follows the stimulation frequency and its harmonics, SSVEPs can be used to build robust high-speed BCI systems to tag perception processes. Up till now, SSVEP-BCI has been the most efficient communication pathway based on non-invasive EEG and has vast potential applications in human augmentation including enhancement of sense, action, and cognitive capabilities[5,7]. To comprehensively study SSVEP characteristics and its potential in BCI control, we devise three separate experiments, including central vision 10 Hz SSVEP, 9-target cued offline BCIs and 40-target cued online spellers.

First, each subject is instructed to focus on a 10 Hz visual flicker in the central visual field for 8 s. All subjects exhibit strong frequency tagging effects at the fundamental frequency ($f_0 = 10$ Hz) and its harmonics based on the proposed SpiralE (the red lines in Fig. 3a, b), thus validating the effectiveness of the in-ear electrodes. More details about the individual spectral representation for all subjects are shown in Supplementary Fig. 13 by the orange lines.

We then conduct the narrow band SNR analysis in a 9-target SSVEP experiment (Supplementary Fig. 14) in which the stimulation frequency ranged from 8-12 Hz in intervals of 0.5 Hz. The group-level SNRs in the in-ear and occipital montages are depicted by the dark and light shades, respectively, in Fig. 3c. The results indicate that a significant harmonic difference exists between two montages in alpha-band SSVEPs, and in-ear SSVEPs exhibit stronger 2nd harmonics with respect to the fundamental frequency. This phenomenon is quite different from the well-demonstrated descending harmonic relationship that are typically observed in the occipital lobe[7]. The different harmonic tendencies are visualized by red and blue lines in the inserted figure in Fig. 3c. In more detail, the harmonic differences between the occipital and in-ear channels across all subjects are plotted in Supplementary Fig. 15. While the fundamental response centralized in primarily occipital lobe, the stronger tendency of in-ear alpha band SSVEP may indicate that the 2nd harmonics are more topographically inclined to occipital-parietal region[40,41]. This result may provide more evidence for studying whole brain dynamics of SSVEP mechanism, which requires further study in a larger demographics.

Furthermore, we compare the narrow band SNR between in-ear (L5, R5), mastoid (M1, M2) and occipital (OZ) montages (Fig. 3d). The comparison of one subject reveal that the spectral intensity of in-ear electrodes is higher than that of the mastoid configuration, which could be attributed to the reduced skull attenuation thanks to the anatomical structure of the ear canal.

For the 9-target SSVEP experiment, a subject-specific calibration algorithm known as the Task-Related Component Analysis (TRCA)[42] (see Methods for more details) has been used to develop a high-performance discrimination model. The classification results of the offline 9-target SSVEP paradigm confirm the feasibility of building an in-ear SSVEP-BCI. On average, 9 subjects achieve an averaged accuracy of 95% in 8 s (blue line in Fig. 3f) and the highest ITR[7] of 36.86 ± 15.53 bits/min (blue line in Fig. 3g) since ear EEG has been reported, further elucidating the efficiency of in-ear SSVEP-BCI based on our SpiralE. The individual classification accuracies of all subjects are shown in Supplementary Fig. 16.

We also implement a 40-target online speller experiment with one subject (Fig. 3e). In contrast to offline experiments, a calibration-free algorithm known as Filter Bank Canonical Correlation Analysis (FBCCA)[43] has been used to provide real-time feedback on the screen in the online in-ear EEG experiments. The online experiments consist of 200 trials, and one subject achieves an overall accuracy of 75.57% in 6 s (red line in Fig. 3f). The peak of ITR achieves 31.24 bits/min at 5.5 s, while the value at 6 s reaches 29.8 bits/min, further demonstrating the feasibility and practicality of using SpiralE to build an online SSVEP-BCI application. After randomized cued spelling, the subject successfully types three phrases, 'THU', 'INEAR', and 'BCI', for demonstration (Supplementary Movie 6). This study implements a 40-target visual BCI based only on in-ear electrodes in a complete calibration-free configuration, thereby verifying the feasibility of developing discreet, wearable and high-performance BCIs based on the proposed SpiralE.

### Auditory BCI based on SpiralE

The AAD problem has been studied mostly in whole-scalp montage where usually 64 or 128 channels are used for accurate decoding. Ear BCI is a natural integration to build intelligent auditory application including attention decoding and EEG-assisted hearing aids in the daily life[44,45]. Thus, in addition to visual BCI, we also demonstrate the efficiency of auditory BCI by AAD in a natural cocktail party scenario with

the proposed SpiralE. In contrast to related studies, in which electrodes are adhered on earplugs or 3D solid supports, the hollow nature of SpiralE fully supports EEG sensing without interfering the auditory sensation.

In our paradigm, the subjects are asked to covertly focus on one of two simultaneously presented auditory stimuli[46] (Fig. 4a). We choose a group of 60 s Mandarin article readings as the stimulation material. The auditory features used for forward and backward modeling are extracted as onset envelopes (black solid line in Fig. 4b), which calculate the salient increase of summed sub-band power of auditory spectrogram (top of Fig. 4b) (see "Methods" for more details). Through forward modeling ("Methods"), we validate the clear attention modulation with SpiralE. The temporal response function (TRF) of one subject shows strong attention effects (Fig. 4c), with the to-be-ignored auditory information mainly filtered via top-down attention control, which is consistent with previous studies[46-48].

Next, we assess the decoding accuracy by backward modeling in a leave-one-out scenario. We introduce separate attended and ignored decoders, and the predicted attention state can be determined by comparing the correlation coefficients between the reconstructed and actual stimulus (Methods), as shown in Fig. 4d. In Fig. 4e, the sample points denote test trials discriminated by attended and ignored decoders for all subjects. The correlation coefficients of the attended or ignored decoder are significant larger in its corresponding class than the other in the group level, demonstrating that the feasibility to identify the attention state using both attended and ignored decoders by SpiralE. The results presented in Fig. 4f shows that the difference between the correlation coefficients (attended minus ignored) is significantly above zero for each subject using the attended model, verifying that SpiralE can be used to distinguish the attended streams from continuous speech in natural scenarios. Then, we compare the classification accuracies based on the signals from scalp, ear (combining the peripheral and in-ear montages), in-ear and peripheral (scalp electrodes at the temporal region) channels (Fig. 4g). The highest accuracy in one subject (S1) can reach 84% where signals based on SpiralE are only used. It is imperative to acknowledge the presence of subject variability in auditory acuity, thus the individual classification accuracies of all subjects are shown in Supplementary Fig. 17. Most importantly, decoding is more effective when the peripheral and in-ear montages are combined (i.e., ear montage), indicating that the proposed SpiralE not only broadens the spatial monitoring range of traditional scalp-based EEG monitoring but also provides a powerful complementary means for effective attention decoding in complex and natural auditory scenarios. Utilizing Pearson's-correlation based algorithm, this study achieves an accuracy rate comparable to the current literature reports for auditory BCI classification using in-ear EEG. It is noteworthy that the potential application of more sophisticated machine learning algorithms, including the introduction of artificial intelligence, could potentially enhance the classification accuracy of auditory BCI in the future.

## Discussion

In summary, the SpiralE developed in this paper can expand and spiral along the auditory meatus under electrothermal actuation, thereby ensuring conformal contact while maintaining communication with the outside world. Moreover, the controlled modulus changes during the deformation, detection and extraction processes reduce friction against the ear canal effectively. Through visual and auditory BCI paradigms, we confirm that the proposed SpiralE can achieve reliable EEG sensing, thus supporting wearable and discreet BCI control. In contrast to the occipital lobe, when using in-ear EEG recordings during alpha-band SSVEP stimulation, we observe noticeable tendencies towards the 2nd harmonic. This suggests that in-ear sensing could be useful for studying the spatial distribution of harmonics in SSVEP research. Besides, we implement 40-target online BCI speller

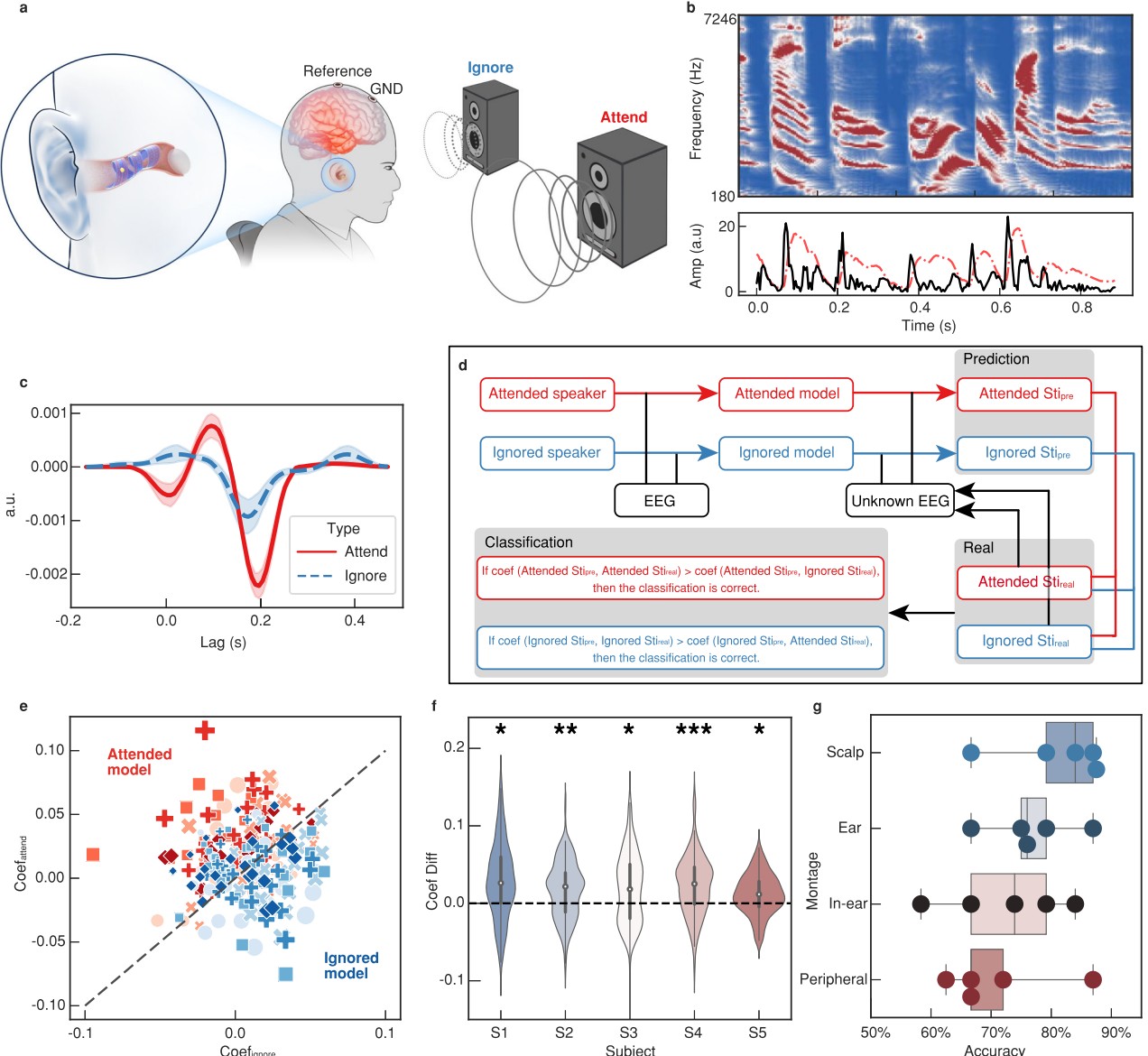

**Fig. 4 | SpiralE for auditory BCI. a** Schematic diagram of the cocktail effect experiment based on SpiralE. **b** The stimulation and envelope extraction. **c** The temporal response functions to attended and ignored speakers in the cocktail effect task ($n = 5$, error bands reflect the 95% confidence intervals of these estimates.). **d** Schematic illustration of the approach to identification of the attended and ignored stimulation. **e** Pearson's correlation coefficients for each subject calculated using the attended and ignored models, with each dot representing different trials of all subjects and the dot size representing the value. **f** The difference between the correlation coefficients of the five subjects ($n = 23, 24, 24, 25,$ and 24 for S1–S5, respectively. Statistic two-sided one sample $t$-test, $p = 1.1e\text{-}2, 1.5e\text{-}3, 3.2e\text{-}2, 5.9e\text{-}5,$ and $1.4e\text{-}2$, respectively. *$p < 0.05$, **$p < 0.01$, ***$p < 0.001$. Boxplot with 25–75th percentiles, mean and whiskers of minima to maxima.). **g** Classification accuracy based on the signals from the scalp, in-ear, peripheral and ear channels ($n = 5$. Statistic two-sided one sample $t$-test. Boxplot with 25–75th percentiles, mean and whiskers of minima to maxima.).

experiments via SpiralE in a complete calibration-free setting, achieving 75.57% classification accuracy. ITR, the most valuable parameter used to evaluate the BCI performance, of the SpiralE in the visual BCI reaches $36.86 \pm 15.53$ bits/min (Supplementary Table 1), demonstrating the feasibility and practicality of using SpiralE to build SSVEP-BCI application. Due to its hollow structure, SpiralE provides an essential platform for decoding covert human attention in natural auditory scenarios, demonstrating that natural and discreet wearable electronics can be integrated into real-life applications. We outlook that the proposed SpiralE-based in-ear BCIs could assist the development of user-friendly biomedical engineering and neuroscience applications in real world.

Ongoing efforts focus on the advancement of in-ear EEG sensing units and the importance of the applications of in-ear BCI paradigms.

Based on the spatial resolution of non-invasive neural signals[49,50], the length of SpiralE, and the paradigms used in BCIs, we have selected five in-ear channels for this study. Future research should conduct a more thorough investigation of the neurophysiological significance of channel selection in ears, to achieve comprehensive coverage of in-ear EEG information in various BCI paradigms while optimizing cost-effectiveness and ensuring robustness across subjects. Besides, the current design of SpiralE places emphasis on a small contact area between the device and the inner wall of ear canal, as well as its hollow features, which may lead to an increased interface impedance due to the lack of external support to ensure contact force. In the future, it would be beneficial to balance the contact force guaranteed by external aids with the contact area and hollow characteristics by exploring alternative driving methods that can allow the device to

actively adapt to the inner wall of ear canal. Moreover, the integration of SpiralE with wireless modules and Augmented Reality glasses will eliminate the limitations of existing rigid and bulky devices and effectively facilitate the development of a flexible and wearable BCIs system with enhanced functionality and user experience.

## Methods

### Description of SMPs
The SMPs used here are thermoset SMPs with reconfigurability of permanent shapes, with which we can effectively adapt the permanent shape of the SMP according to the mold[51–53]. The molecular structure and polymer network of the two kinds of SMPs involved are shown in Supplementary Fig. 2. SMP1 and SMP2 have similar structures, but the molecular weights of polycaprolactone diol (PCL) in SMP1 and SMP2 are different. The molecular weight of PCL in SMP1 is 10,000 g/mol, while it is 2000 g/mol in SMP2. The differential scanning calorimetry (DSC) and dynamic mechanical analysis (DMA) curves of the two materials are shown in Supplementary Fig. 3, indicating that the melt transition temperatures ($T_{trans}$) are approximately 50 °C and 37 °C, respectively. The conversion in moduli (Supplementary Fig. 3b) not only reduces risk of damage to ear canal during deformation, and thus achieving a conformal fit with the inner wall of canal but also ensures a reliable support of the deformed SpiralE, thereby realizing a tight fit during in vivo experiments. In addition, the biocompatibility of these materials was demonstrated previously[33].

### Fabrication of SpiralE
First, the silicon wafer, which served as a hard, flat base, was cleaned with acetone, anhydrous ethanol, and deionized (DI) water in sequence. Then, the surface was blown dry with nitrogen and heated at 110 °C for 5 min for further drying. Next, a layer of polymethylmethacrylate (PMMA) was spin-coated on the substrate, followed by heating as a sacrificial layer. Then, polyamic acid solution was spin-coated on the PMMA surface, followed by step heating to obtain the flexible PI substrate. Cr (10 nm)/Au (150 nm) and Cr (10 nm)/Au (300 nm) layers were evaporated on the PI substrate via electron beam evaporation, and the pattern of the EAL and the underlying pattern of EEGDL were obtained by a standard photolithography process. For the EEGDL, the PI layer protected the interconnected lines in the top layer. Then, a Cu layer was evaporated and photolithographed to obtain the top layer PI mask using another set of masks by exposing five metal sites. The EEGDL and EAL were prepared (Supplementary Figs. 4a1 and 4a2), respectively.

After the first transfer printing step, two layers were printed on the flexible PDMS substrate. Then, we patterned PI films by reactive ion etching (RIE) to obtain stretchable parts (Supplementary Fig. 4b). The two stretchable parts were integrated with the corresponding SMP substrates through the second transfer printing step (Supplementary Fig. 4c).

During this process, the EEGDL was transferred to the surface of SMP2 (Supplementary Fig. 4c1), and the EAL was transferred to the surface of SMP1 (Supplementary Fig. 4c2). Then, the side of SMP2 without the EEGDL was attached to the side of SMP1 with the EAL to construct the planar electronics, as shown in Supplementary Fig. 4d, the inset of which showed a schematic diagram of the stacking of EEGDL and EAL. We note that the structure does not delaminate during the deformation and signal acquisition processes[32].

After forming the planar structure, we constructed the three-dimensional electronics. First, the electronics was twined on and fixed on a cylinder mold with a diameter of 5.5 mm ($2r_0$) (Supplementary Fig. 4e). Here, $r_0$ was designed to be slightly larger than the empirical value of the inner radius of the ear canal. Then, we placed the electronics in an oven at 175 °C for 15 min to reconfigure its permanent shape to pre-designed form. Next, the structure was cooled to room temperature to obtain the permanent form of the electronics

(Supplementary Fig. 4f). Then, we placed the electronics in hot water (50 °C) to soften it and wrapped the electronics around another cylinder with a diameter of 3.6 mm ($2r_1$) (Supplementary Fig. 4g) to obtain the temporary small spiral configuration (Supplementary Fig. 4h). Thus, the permanent and temporary shapes of SpiralE, namely, a large spiral with a diameter of 5.5 mm and a small spiral with a diameter of 3.6 mm were obtained, respectively.

It is noteworthy that the fusion of SpiralE with the external processor has been accomplished via a sequence of flexible anisotropic conductive film, commonly referred to as ACF, customized printed circuit board (PCB), DuPont wires, and EEG crocodile clip cables. Specifically, the ACF-facilitated thermal bonding process has been employed to secure the EAL and EEGDL connected to the tailored PCB, subsequently allowing for the establishment of interconnects between the EAL and a DC power source, as well as between the EEGDL and an EEG amplifier via DuPont wires and alligator clips (Supplementary Fig. 9).

### Transfer printing process
The first step in the transfer printing process was using an elastic stamp to quickly pick up the layers from the Si substrate and then print them slowly on the PDMS substrate. Similarly, the second step in the transfer printing process quickly picked up the layers from the PDMS substrate and printed them slowly on the SMP substrates.

### Electrochemical characterization
Electrochemical experiments were carried out on a CS350-type electrochemical workstation (CorrTest Instruments, China). The three-electrode configuration was used with SpiralE, platinum, and silver electrodes serving as the working electrode, counter electrode, and reference electrode. The sweep frequency ranged from 0.1 Hz to 10 kHz at room temperature (27 °C).

### Equipment and procedure for BCIs
The equipment used in the BCI experiments included a commercial EEG recorder (Synamps2, NeuroScan, consisting of a 64-channel EEG cap), SpiralE, and two wet commercial electrodes that were placed around the mastoid for comparison (M1, M2). The SpiralE defined L1 - L5 and R1 - R5 as the left and right ears, respectively. All recorded channels used a common reference (REF) and ground (GND) in the 10–20 system. The visual stimulations were presented on an LCD screen (Acer GD245 HQ, 60 Hz refresh rate), and the natural speeches in the auditory decoding experiments were played via two separate audio speakers (EDIFIER R101V 2.1).

Prior to the experiment, a comprehensive explanation of the experimental protocol was provided to all participants, and they signed the informed consent statement. As part of our study protocol, we afforded participants with the autonomy to withdraw from the experiment at any point in accordance with their own preferences, without incurring any untoward consequences.

After that, the auditory meatus of each subject was cleaned with medical alcohol to remove grease and earwax. Then SpiralE was inserted into ear in the small spiral configuration. A DC voltage of -10 V was applied to electrothermally actuate the expansion and deformation processes. In this process, participants were equipped with the ability to modulate the temperature elevation through manipulation of the voltage knob, thereby enabling them to terminate the experiment at their discretion. During this process, an ear endoscope (Bebrid Note3) was used to inspect the deformation and assisted SpiralE to fit snugly into the ear canal. In case the impedance value still cannot be lowered into the acceptable range (≤50 kilohms), the SpiralE was taken out and reshaped into a small spiral. A small amount of gel was applied to SpiralE, and it is then reinserted into the ear canal and actuated again. Out of the 18 SpiralEs (9 subjects × 2 SpiralEs) involved in the visual experiments, a total of 10 had gel applied. These particular

SpiralEs can be identified by their blue highlighting in Supplementary Table 2. When the power was removed, SpiralE was self-supported in ear canal after cooling to room temperature. Similarly, a voltage was applied to SpiralE after the end of the experiment. After the modulus decreased, SpiralE was carefully removed and cleaned using cotton swabs and water for reuse by next subject. The subjects were also instructed to clean their external auditory meatus using cotton swabs and alcohol to ensure proper hygiene. Notably, all participants did not report any discomfort throughout the ~2 h experiment, irrespective of the presence or absence of gel.

## BCI experiments setting

Sex was considered in the study design and both males and females were recruited. Participants were solicited to self-report their biological sex. Specifically, nine subjects (7 males and 2 females) with normal vision and an average age of 26.89 years (s.d. 3.1) participated in the visual experiments, and 5 subjects (4 males and 1 female) with normal hearing and an average age of 25 years (s.d. 1.7) participated in the auditory experiments. The experimental techniques and procedures employed in this study were not expected to elicit any deleterious effects, whether physical, psychological, social, or otherwise, on the subjects. Owing to the non-invasive and well-tolerated nature of the experimental procedures, all participants were able to successfully complete BCIs without experiencing any discomfort or complications. All measurements were approved by the Institution Review Board of Tsinghua University (20220023). During EEG acquisition, subjects were asked to sit comfortably in a shielded room while maintaining a distance of 65 cm from the screen and speakers.

In the visual BCI experiments, the subjects were asked to open their eyes for 1 min and close their eyes for 1 min before SSVEP stimulation. First, a central flicker with 10 Hz SSVEP modulation was presented for 20 8-s trials. Then, 9 targets with stimulation frequencies ranging from 8 Hz to 12 Hz in intervals of 0.5 Hz were displayed for 12 blocks (12 blocks * 9 targets = 108 trials) according to the layout. Before each trial, a visual cue was randomly assigned for each target (Supplementary Fig. 14), and the subject was required to overtly attend the cued target without blinks or movement. We chose one subject for the 40-target online SSVEP speller experiment[7], in which the online discrimination character was presented on the screen in real time. The cued spelling process included 5 blocks (200 trials in total) and was initiated without calibration. Each target identification result was reported using a 6 s in-ear EEG. After 200 trials of randomly cued targets, the subject was asked to type 'THU', 'INEAR', and 'BCI', and the video footage was shown in Supplementary Movie 6. The consent to publish has been obtained by the subject in the video.

For the auditory decoding experiments, the paradigm replicated the design[46], in which two audio materials were streamed simultaneously on two separate speakers. Subjects were required to covertly focus on only one speaker and completed reading comprehension questions related to the attended material.

## Signal processing and decoding algorithm

For the visual BCI experiments, the raw EEG data were bandpass filtered from 0.5 to 90 Hz, while for the auditory BCI experiments, the data were processed with an anti-aliasing filter from 2 to 8 Hz. The visual and auditory data were downsampled to 250 Hz and 128 Hz, respectively. Data preprocessing also included channel and epoch cleaning. Bad epochs and channels were interpolated and rejected automatically using the autoreject package[54]. Bad epochs were rejected if over 4 out of 10 channels were considered invalid; if not, the invalid channels were interpolated based on the remaining ones.

This study included three decoding algorithms with different aims. Two coefficient-based spatial filter techniques were used for visual decoding. Spatial filtering methods used a set of spatial filters $W_x$ and $W_z$ to optimize the correlation between the signals and templates

as follows:

$$\rho^{(f)} = \mathrm{corr}\left(W_Z^{\mathrm{T}} Z^{(f)}, W_X^{\mathrm{T}} X\right) \tag{1}$$

where $X$ stands for the to-be-classified signal and $Z^{(f)}$ denotes the $f_{\mathrm{th}}$ template.

The final result was determined as the target corresponding to the largest coefficient. For the 9-target SSVEP-BCI experiment, we used the TRCA algorithm for classification[42]. The TRCA algorithm takes trial-averaged training data as a template, and the spatial filter is computed by maximizing the task components across the training data. We used cross validation to evaluate the performance; in each validation, one block of testing data was evaluated by the model fitted by 5 blocks of training data. We then used the calibration-free FBCCA algorithm for 40-target online decoding[43]. Unlike the TRCA method, the FBCCA algorithm takes a standard sine/cosine signal as a template and thus does not rely on training data[42]. All 5 blocks of EEG data, which included 200 trials, were used for performance evaluation. We first calculate the narrow band SNR spectrum for 9-target SSVEP[55]. Here the noise is considered to be the power spectra density around neighboring $K$ frequencies on both sides.

$$\mathrm{SNR} = \frac{K \times F(f)}{\sum_{k=1}^{K/2}\left[F(f + k\Delta f) + F(f - k\Delta f)\right]} \tag{2}$$

where $f$ is the stimulation frequency and $K$ denotes the number of neighboring frequencies ($K = 4$ in this study). Finally, the BCI performance was evaluated by ITR[7]:

$$\mathrm{ITR} = \left(\log_2 M + P\log_2 P + (1 - P)\log_2\left(\frac{1 - P}{M - 1}\right)\right)/T \tag{3}$$

where $M$ is the number of classes, $P$ is the accuracy, and $T$ is the time used for classification (in second).

The AAD can be solved by building a linear regression model between the stimulus signals and EEG responses:

$$x(t,n) = \sum_\tau w(\tau,n)s(t - \tau) + \varepsilon(t,n) \tag{4}$$

Thus, the EEG signal $x(t)$ can be modeled according to the convolution between the linear weight $\omega(\tau,n)$ termed the TRF and the stimulus $s(t)$[56], which is known as forward modeling. Moreover, the stimulus can be reconstructed by reversing the convolution relationship, which is known as backward modeling. To represent stimulus $s(t)$, each auditory signal was converted to spectrogram using 128 sub-band filters range from 180 to 7246 Hz[57]. The spectrogram was then summed up across 128 bands to produce broadband envelope (red dashed line in Fig. 4b). Finally, the onset envelope was calculated by taking the first derivative of the broadband envelope. The TRF and the discrimination problem were implemented by eelbrain[58]. For the backward classification, we followed the procedure[46], which involved calculating the correlation coefficient between the reconstructed and real stimuli. After bad trials were rejected, 23, 24, 24, 25, and 24 trials remained for S1−S5, respectively. The evaluation was conducted according to a leave-one-out design, where one data trial was used for discrimination and the remaining data were used for modeling.

## Reporting summary

Further information on research design is available in the Nature Portfolio Reporting Summary linked to this article.

## Data availability

The collected visual and auditory experiments are provided via Zenodo (https://doi.org/10.5281/zenodo.7748035). A detailed manual

for the data analysis and deployment can be found in the Zenodo repository. Raw EEG data can be obtained from corresponding authors on reasonable request. Source data are provided with this paper.

## Code availability

The codes that support this study, including the offline resting state and SSVEPs are available from Zenodo (https://doi.org/10.5281/zenodo.7748035).

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

## Acknowledgements

This work was supported by the National Natural Science Foundation of China (No. U20A6001 and 11921002, X.F.). We sincerely thank all subjects who participated in the experiments in this study.

## Author contributions

X.F., Y.M., X.G., Z.W., and N.S. conceived and designed the project. Z.W. fabricated the device. Z.W. and N.S. conceived and performed in-ear EEG experiments. N.S. performed data analysis. N.Z. and T.X. synthesized the SMPs. N.Z., Y.J., and Y.T.W. assisted in characterization of the SMPs. H.L. and J.C. assisted in fabrication and in vitro characterization of the device. X.Z. assisted in the observation of the device in ears. Z.W., N.S., Y.Z., and H.L. wrote the manuscript and Y.H.C., Y.C., H.W., Y.J.W., Y.M., X.G., and X.F. provided feedback on the manuscript.

## Competing interests

The authors declare no competing interests.
