## [Peer Review File · Nature Communications]

REVIEWER COMMENTS

Reviewer #1 (Remarks to the Author):

Abstract:

- Abstract only mentions cap and microneedle-based invasive approaches. What about headband or wearable BCIs? Other in-ear works also exist.

- What is active deformation? Is there a motor or microfluidic channel? Otherwise, isn't that just 'passive deformation'?

Line 38: What about wearable EEG BCIs such as the Muse headband?

Line 53 - 57: While I agree with points 1 and 3 - many of your references actually have visibly hollow in-ear eeg earpieces. It seems like these do not occlude sound or hurt communication. Rather, it seems like the user-generic options (Kaveh 2020, Kidmose 2013, Schwendeman 2022, Bertelsen 2019) all stay close to the entrance of the ear canal (likely due to subject-subject ear canal variations). This concept of a spiral, deformable electrode is alluded to in point 3, but is not put in terms of electrode contact area - which is the technical benefit here (at least from my perspective). Links to the papers below:

https://ieeexplore.ieee.org/abstract/document/8857351?casa_token=FUeso13L2GUAAAAA:RI2wWYtnwto8-V7_LAJtsJghHR7xVXmtDm9MO2PXrz3HM4ScPXSEkm-l7NJPtFvPxMPfmQ

https://ieeexplore.ieee.org/abstract/document/9871859?casa_token=rttpw1adbSGcAAAAA:5_pEzYPLxVWYlc5C931sRrAFd-nVcuRLmyUBLYCjn76SM75AchUeiaPUaTuulyKOHAsjgQ

https://ieeexplore.ieee.org/abstract/document/6609557?casa_token=htWCLiwmOp8AAAAA:0kJV8WTI-zdTSXQG0E-5wfzec4r78J0FelxMqzm_I01OITQ7gMX8qDSbw0icQDR0zDox8A

https://ieeexplore.ieee.org/abstract/document/9115876?casa_token=DxLEiIVRg3MAAAAA:PW2W9eajChQArgemvd0tSuX_yOeKWtTdw61myOyHJvb2qpFPy65Eh583sohGDB2OZf8o8w

line 63: Again "active deformation" - so far you haven't defined it. A spiral is just a coiled spring - that doesn't seem very active to me.

line 66: is it really a smaller contact area? Do you compare contact areas to other works?

line 80: Overall, these results are exciting. From what I could see, other ear eeg works have only been demonstrated with simpler experiments.

line 89: Grammar error: "SpiralE can conform" not conformal

page 5 in general:

Discussions of deformation and thin films should be followed by fatigue or lifetime measurements. Do you foresee this as a single-use, short-term use (a handful of measurements), or long-term use (year+)?

Line 103 again alludes to the fact that this does not occlude hearing, but the earlier references also do not occlude hearing. Unfortunately, a hollow assembly is not pushing the state of the art anymore.

Line 146: I request you also report the impedance at 50 Hz (or something that is inside the EEG band of interest).

Line 191: you refer to these as active electrodes but I don't see any amplifiers near the electrodes. Are they really active electrodes? Furthermore, there is still no description of "active deformation". It seems like the spiral is just using the built in spring constant from the SMP layers (and the whole assembly I suppose). Furthermore, what are the areas of the electrodes? Impedance of an electrode isn't that helpful without the area. Furthermore, it is important to report the frequency at which you measure impedance (<50kOhms at what frequency?).

Figures 3 & 4 are beautiful. I would suggest making the text slightly larger but that is nitpicky.

Overall, this paper presents a novel in-ear electrode architecture (a spiral vs. traditional earplugs or ear molds) as well as multiple BCI relevant experiments to showcase it's application. This spiral architecture is user-generic but requires hydrogel and skin cleaning (this was hidden in the methods and should be clearly stated in the results - especially when comparing oneself to dry-electrode ear eeg works). Unfortunately, it is unclear how practical this sort of electrode architecture may be in real-world scenarios (hearing aids, day-to-day BCIs, etc.) - how often can it be re-used? Do hydrogels need to be reapplied?... These sorts of questions are left unanswered. While the discussion claims that this work

introduces a new communication channel for overt visual and covert auditory BCIs, I don't quite see if that is a major claim. This work did not invent the experiment or the channel (ear eeg seems to have been around for over a decade). They do, however, demonstrate BCI experiments with greater fidelity and accuracy than I've seen for ear eeg works. Regardless, the main novelty of this work is the electrode technology. As a result, it would be beneficial for a clear discussion/comparison versus other electrode technologies (maybe discuss area vs. impedance or compare to conventional wet electrodes).

Lastly, this work makes certain claims that are misleading, these electrodes are not "actively deformed", they are deformed and then allowed to expand inside a closed space. This is a coiled spring that they are releasing. Active deformation implies a driving force (like a motor or microfluidics). Secondly, these are not active electrodes. From my understanding, active electrodes require electrode amplifiers right beside the electrode interface in order to minimize the effects of motion and wireline interference.

Unfortunately, I cannot recommend this for publication in nature due to its smaller scope and lack of practicality. This does not present an entire system, it really presents just an electrode technology (may be a better fit for advanced materials or sensors). That being said, the methodology and figures are all sound/beautiful. It really is a great work. I just don't think it's right for nature.

Reviewer #2 (Remarks to the Author):

Summary:

Z. Wang et al. reported an in-ear bioelectronic device, SpiralE, which can be conformally integrated with auditory meatus. This can be achieved by active thermal deformation of encapsulating shape-memory polymers, enabling facile insertion and adaptive expansion to achieve conformal contact in a spiral along with the inner wall of ear canal. The stretchable EEG electrodes as well as an electrothermal actuator offered mechanical reliability for the in-ear integration. Using the SpiralE, the authors demonstrated in-ear EEG detection, which can be used in SSVEP BCI classification and 40-target online speller experiments as well. Because these concepts, analysis, and demonstrations are impressive, I recommend this work to be published in Nature Communications after addressing minor issues. Some detailed comments that can be helpful for authors to improve this manuscript are as follows:

Comments:

Comment #1: A major issue of wearable electronics is a bulky external processor for recording and processing EEG data, although the bioelectronic device is soft and thin. These processors are difficult to be integrated with the body, causing discomfort and irritation. Please discuss the integration strategies of in-ear bioelectronics with the external processor.

Comment #2: Another issue is the interconnections between the flexible device and the external processor. The wirings can be easily disconnected, limiting dynamic physical activities. Please describe how the authors interconnected the SpiraleE to the external measurement setup and discuss its reliability for continuous in-ear EEG recording.

Comment #3: In my opinion, SpiraleE may be stiff because it maintained the spiral shape after conformally implanted in the inner wall of the ear canal. However, the wearable device should be soft to avoid irritation caused by dynamic human motion. In this regard, please provide the mechanical properties (e.g., modulus or stiffness) of SpiraleE at each deformation stage to estimate the device's softness.

Comment #4: The authors analyzed the impedance of EEG electrode and claimed that all values were within the acceptable range. Then, is the impedance of EEG electrode still sufficiently low even if EEG is measured without a gel lowering the impedance? Because it seems difficult to insert the gel between the electrode and the inner wall of ear canal that are in conformal contact, the impedance should be low even without gel.

Comment #5: It would be helpful to provide detailed information on SMP1 and SMP2 (e.g., molecular structure and polymer network) for better understanding.

Comment #6: The authors claimed that 'ESCL is the SMP2 layer'. What is ESCL?

Reviewer #3 (Remarks to the Author):

The authors introduce a conformal in-ear EEG sensor named SpiraleE for visual and auditory brain-computer interface (BCI). Joule heating can expand the spiral-shaped electrodes supported by a shape memory polymer (SMP), conforming to various ear canal shapes. Because the electrode does not require rigid supports, such as earphones or ear plugs, the user's auditory perception is unobstructed.

The reported device has novelty, and the authors successfully demonstrated its potential as BCI. Videos and supplementary materials are very helpful for understanding the work. The authors may consider the following points to further improve the manuscript.

1. The mechanical properties of SpiraleE, such as its thickness and modulus, should be explicitly reported to support the ear conformability of the electronics. Also, the ear canal is not perfectly cylindrical, and there can be localized curvatures inside. Please comment on whether the SpiraleE is soft enough to conform to nonuniform curvatures.
2. Fig. 2a and 2b show the impedance spectroscopy of the device itself, not an electrode-skin contact impedance. Although the figure illustrates that the impedance (primarily resistance in this case) is not affected by expansion, it cannot fully verify the stability of the device as bioelectronics. Supplementary Fig. 3 shows the electrode-skin impedance recorded by an EEG device, but biogel was applied for this measurement, diminishing its advantages as a dry thin-film-based electrode. The authors should perform the electrode-skin contact impedance spectroscopy without gel to make a fair comparison with conventional EEG systems and other dry electrodes in the literature.
3. Since the clear in-ear montage is not provided, there is no clear justification for having five in-ear channels. As a good BCI carefully selects channels near the source of the EEG modulation to a specific BCI task, authors should provide the proper neuroscientific reasoning, if any, behind the design and channel selection of SpiraleE.
4. The authors use signal-to-noise ratio (SNR) as a validation metric, but no clear definitions of “signal” and “noise” are provided. As there is no generic definition of signals and noises for EEG, the authors should provide their own definitions and reasoning behind these definitions.
5. Fig. 4e shows that the Pearson’s-correlation-based temporal response function (TRF) models may not offer the best classification since many dots are placed across the boundary. The authors may perform feature analysis or try other machine-learning approaches to further improve the classification.

Responses to reviewers' comments

Reviewer #1

Comment 1

Abstract:

- Abstract only mentions cap and microneedle-based invasive approaches. What about headband or wearable BCIs? Other in-ear works also exist.
- What is active deformation? Is there a motor or microfluidic channel? Otherwise, isn't that just 'passive deformation'?

Our response:

We are grateful for the reviewer's insightful comments and would like to extend our apologies for any confusion that may have arisen as a result of our previous statements.

We acknowledge that our previous abstract may have been somewhat one-sided, as it only mentioned the most widely recognized cap and microneedle-based invasive BCIs. In view of your comments, we have revised this section of the abstract to be more balanced and inclusive. A more

detailed explanation of the headband is listed in the following response. Additionally, we have comprehensively listed and compared other in-ear works in Table r1 (Supplementary Table 1) of the revised manuscript to highlight the distinct advantages of SpiraleE.

In addition, the “active deformation” mentioned in the original article does contain ambiguous language, which could lead to misinterpretation. This phrase refers to SpiraleE’s capability to adaptively conform to the inner wall of the external auditory canal after SMPs are actuated by the electrothermal drive layer. In essence, we integrate an electrothermal actuation layer with SMPs and an EEG detection layer to achieve a unified system of drive, deformation, and EEG monitoring, which is also a notable innovation of SpiraleE. In view of your comments, we have deleted the phrase “*via active deformation*” in the title and made appropriate changes to describe the deformation mechanism of SpiraleE at the corresponding position in the modified manuscript.

Table r1. Comparison of recent ear EEG electronics.

		Ref ¹	Ref ²	Ref ³	Ref ⁴	Ref ⁵	Ref ⁶	Ref ⁷	Ref ⁸	Ref ⁹	This study
Electrodes Properties	Location	In-ear	In-ear	Auricle, mastoid	In-ear	Behind ear	Auricle, in-ear	In-ear	Auricle, in-ear	Auricle, in-ear	In-ear
	Support	Earplug	Earphone	-	Earplug	Cap	Earmould silicone	Earmould silicone	Resin (PC/PI)	Resin (PC/PI)	SMPs
	Contact Area	Whole canal	Whole canal	-	Whole canal	Full head	Whole canal	Whole canal	Whole canal	Whole canal	Spiral
	Electrode Area	-	-	-	-	-	9.6 mm ²	-	60 mm ²	-	0.785 mm ²
	Electrode Material	Ag	CNT/PDMS	Au	Wet Ag coated nylon	Ag/AgCl	Ti/IrO ₂	IrO ₂	Ag	Ag	Au
	Skin Impedance (50 Hz)	5 k	~150 k	-	10 k	-	435 k	-	392 k	190 k	111 k
	Sensors	8	1	5	2	8	8	12	6	10	10
Features	Style	Generic	Generic	Generic	Generic	Generic	Customized	Customized	Generic	Generic	Generic
	Communication	No	No	-	No	-	No	No	Yes	Yes	Yes
	Structure	Stuffed	Stuffed	-	Stuffed	-	Stuffed	Stuffed	Hollow	Hollow	Hollow
Experiments	Visual	SSVEP	SSVEP	2-target SSVEP 36-target P300: 12 bpm	-	12-target SSVEP: 30.21 ± 10.61 bpm	SSVEP	-	-	-	9-target SSVEP 40-target speller: 36.86 ± 15.53 bpm
	Auditory	ASSR	AEP/ASSR	-	ASSR	-	ASSR	ASSR	ASSR	-	AAD: Accuracy : 72.42%
	Other	Alpha rhythm	Alpha rhythm	Alpha rhythm	-	-	Alpha rhythm	-	Alpha rhythm	Drowsiness detection	Alpha rhythm
	Subjects	14	6	3	5	16	12	10	3	5	14

Modification to the manuscript:

On lines 23-25, page 1, we changed the sentence to: “Most devices use cap-based noninvasive, headband-based commercial products or microneedle-based invasive approaches, which are constrained for inconvenience, limited applications, inflammation risks and even irreversible damage to soft tissues.”

The title has been replaced with “Conformal in-ear bioelectronics for visual and auditory brain-computer interfaces”. Additionally, we have replaced the phrase “*via active deformation*” with “*under electrothermal actuation*” at the corresponding place in the modified text, which accurately describes the deformation mechanism of Spirale.

References:

- 1 Kidmose, P., Looney, D., Ungstrup, M., Rank, M. L. & Mandic, D. P. A study of evoked potentials from ear-EEG. *IEEE Trans. Biomed. Eng.* **60**, 2824-2830, (2013).
- 2 Hoon Lee, J. *et al.* CNT/PDMS-based canal-typed ear electrodes for inconspicuous EEG recording. *J. Neural Eng.* **11**, 046014, (2014).
- 3 Norton, J. J. S. *et al.* Soft, curved electrode systems capable of integration on the auricle as a persistent brain-computer interface. *Proc. Natl. Acad. Sci. U. S. A.* **112**, 3920-3925, (2015).
- 4 Goverdovsky, V., Looney, D., Kidmose, P. & Mandic, D. P. In-Ear EEG from viscoelastic generic earpieces: robust and unobtrusive 24/7 monitoring. *IEEE Sens. J.* **16**, 271-277, (2016).
- 5 Wang, Y. T. *et al.* An online brain-computer interface based on SSVEPs measured from non-hair-bearing areas. *IEEE Trans. Neural Syst. Rehabil. Eng.* **25**, 14-21, (2017).
- 6 Kappel, S. L., Rank, M. L., Toft, H. O., Andersen, M. & Kidmose, P. Dry-contact electrode ear-EEG. *IEEE Trans. Biomed. Eng.* **66**, 150-158, (2019).
- 7 Bertelsen, A. R. *et al.* Generic dry-contact ear-EEG. // 2019 41st Annual International Conference of the IEEE Engineering in Medicine and Biology Society (EMBC), 5552-5555, (2019).
- 8 Kaveh, R. *et al.* Wireless user-generic ear EEG. *IEEE Trans. Biomed. Circuits Syst.* **14**, 727-737, (2020).
- 9 Schwendeman, C., Kaveh, R. & Muller, R. Drowsiness detection with wireless, user-

generic, dry electrode ear EEG. // 2022 44th Annual International Conference of the IEEE Engineering in Medicine & Biology Society (EMBC), 9-12, (2022).

Comment 2

Line 38: What about wearable EEG BCIs such as the Muse headband?

Our response:

We would like to express our appreciation for the valuable comment provided by the reviewer, and we apologize for any confusion that may have resulted from our previous manuscript.

As stated, headbands integrated with a few electrodes are commonly utilized for daily EEG monitoring due to their gel-free nature, which allows for safe and easy application to non-hair areas. However, it is important to note that headband EEG sensors, such as the Muse band, have inherent limitations. These sensors can mainly capture neural activity from the frontal lobe, and thus are marketed primarily as tools to facilitate meditation and reduce stress. While the frontal lobe plays a crucial role in top-down control and is associated with various high-level physiological effects, the causal relationship between these factors and low-resolution modalities such as EEG is still not fully established. Therefore, developing frontal lobe BCI for control purposes remains challenging.

Our proposed in-ear bioelectronics, on the other hand, can capture visual information from the occipital lobe and auditory information from the temporal lobe, which may be decoded for external control. This novel approach opens up exciting possibilities for more robust applications beyond meditation and stress reduction. In summary, we recognize that headband and ear EEG electronics are designed for different purposes, and we believe that ear EEG BCI has more solid applications beyond just helping with meditation. In the modified version, we included a wearable headband EEG system for comparison.

If we extend the discussion slightly, considering the aesthetics of wearable EEG electronics, hand bands have been proposed for years but have not been fully accepted by the customer market. The reason may be that people are reluctant to put a band on their head, but in contrast, it is easy to integrate sensors into the ear area (plug/headphone).

Modification to the manuscript:

On lines 38-41, page 2, we changed the sentence to: “**However, most BCIs involve cap-based noninvasive methods^{1,2}, headband-based commercial products^{3,4}, or microneedle-based invasive**

methods⁵⁻⁹, which are not acceptable for daily use due to inconvenience and inflammation risks and even irreversible damage to soft tissues.”

References:

- 1 Chen, X. *et al.* High-speed spelling with a noninvasive brain-computer interface. *Proc. Natl. Acad. Sci. U. S. A.* **112**, E6058-E6067, (2015).
- 2 Penaloza, C. I. & Nishio, S. BMI control of a third arm for multitasking. *Sci. Robot.* **3**, eaat1228, (2018).
- 3 Hashemi, A. *et al.* Characterizing population EEG dynamics throughout adulthood. *ENeuro* **3**, 0275-16, (2016).
- 4 Arsalan, A., Majid, M., Butt, A. R., & Anwar, S. M. Classification of perceived mental stress using a commercially available EEG headband. *IEEE J. Biomed. Health Inform.* **23**, 2257-2264, (2019).
- 5 Bouton, C. E. *et al.* Restoring cortical control of functional movement in a human with quadriplegia. *Nature* **533**, 247-250, (2016).
- 6 Ajiboye, A. B. *et al.* Restoration of reaching and grasping movements through brain-controlled muscle stimulation in a person with tetraplegia: a proof-of-concept demonstration. *The Lancet* **389**, 1821-1830, (2017).
- 7 Pandarinath, C. *et al.* High performance communication by people with paralysis using an intracortical brain-computer interface. *eLife* **6**, e18554, (2017).
- 8 Musk, E. An integrated brain-machine interface platform with thousands of channels. *J. Med. Internet Res.* **21**, e16194, (2019).
- 9 Chaudhary, U. *et al.* Spelling interface using intracortical signals in a completely locked-in patient enabled via auditory neurofeedback training. *Nat. Commun.* **13**, 1236, (2022).

Comment 3

Line 53 - 57: While I agree with points 1 and 3 - many of your references actually have visibly hollow in-ear eeg earpieces. It seems like these do not occlude sound or hurt communication. Rather, it seems like the user-generic options (Kaveh 2020, Kidmose 2013, Schwendeman 2022, Bertelsen 2019) all stay close to the entrance of the ear canal (likely due to subject-subject ear canal variations). This concept of a spiral, deformable electrode is alluded to in point 3, but is not

put in terms of electrode contact area - which is the technical benefit here (at least from my perspective).

Our response:

We appreciate your suggestion regarding SprialE's ability to avoid occluding hearing. We conducted a comprehensive review of the literature (Table r1, also known as Supplementary Table 1), including the sources you suggested. The ability to avoid occluding hearing is not unique to the SprialE. Kaveh¹, Schwendeman² have also utilized a hollow structure similar to SprialE in their research. Although Kaveh's study is undoubtedly inspiring, we have found that the 3D-molded support structure, despite its hollow and user-generic design, lacks the adjustability and adaptability necessary to conform to individual canal shapes. We posit that this inflexible and solid design may lead to a compromised user experience.

Other works, although being hollow, can still impede auditory perception. For example, Kidmose³ drilled a little hole in the support structure to play ASSR stimulus. We suspect that this kind of design will much likely to suffer from great loss of environmental voice, so that the related studies^{3,4} can only perform experiments using earbuds. In contrast, during our experiments, the natural auditory stimulus is played using microphone speakers, which fully simulated the actual auditory perception in the real life.

Consequently, we propose that our soft bioelectronics design is better suited to leverage the benefits of hollowness, accommodate inter-subject variability, and enhance user experience, making it a more suitable option for everyday usage.

Additionally, we would like to express our appreciation for the constructive feedback provided by the reviewer regarding the electrode contact area in our study. We acknowledge that we overlooked an important aspect of highlighting the contact area, and we thank the reviewer for bringing this to our attention. It is worth noting that in most other studies, the entire ear canal is covered to achieve conformal contact, which can lead to device failure and damage to the ear canal. As emphasized by the reviewer's comments, SprialE offers the additional advantage of monitoring in-ear EEG with a smaller contact area, in addition to its switchable modulus, thereby reducing the risk of scratching the inner wall of the ear canal. We have included additional information on the contact area and electrode area in the revised manuscript in response to the reviewer's suggestion.

Modification to the manuscript:

On lines 112-116, page 6, we added the following sentences, "In addition, SprialE has been

designed as a long strip with an in-plane size of 50 mm * 3 mm (where the electrode is a circle with a diameter of 1 mm) and is supported on the ear canal with a spiral shape to finish in-ear EEG recording, rather than the previous devices that have full contact with the inner wall of the ear canal (Supplementary Table 1). This strategy further reduces abrasion and friction with the ear canal.”

References:

- 1 Kaveh, R. *et al.* Wireless user-generic ear EEG. *IEEE Trans. Biomed. Circuits Syst.* **14**, 727-737, (2020).
- 2 Schwendeman, C., Kaveh, R. & Muller, R. Drowsiness detection with wireless, user-generic, dry electrode ear EEG. // *2022 44th Annual International Conference of the IEEE Engineering in Medicine & Biology Society (EMBC)*, 9-12, (2022).
- 3 Kidmose, P., Looney, D., Jochumsen, L., & Mandic, D. P. Ear-EEG from generic earpieces: A feasibility study. // *2013 35th annual international conference of the IEEE engineering in medicine and biology society (EMBC)*, 543-546, (2013).
- 4 Kidmose, P., Looney, D., Ungstrup, M., Rank, M. L. & Mandic, D. P. A study of evoked potentials from ear-EEG. *IEEE Trans. Biomed. Eng.* **60**, 2824-2830, (2013).

Comment 4

Line 63: Again "active deformation" - so far you haven't defined it. A spiral is just a coiled spring - that doesn't seem very active to me.

Our response:

We express our gratitude for the valuable comments provided by the reviewer and humbly offer our apologies for any perplexity that may have resulted from our previous statements. As mentioned in response to comment 1, the “active deformation” mentioned in the original article refers to Spirale’s capability to adaptively conform to the inner wall of the external auditory canal after SMPs are actuated by the electrothermal drive layer. In essence, we integrate an electrothermal actuation layer with SMPs and an EEG detection layer to achieve a unified system of drive, deformation, and EEG monitoring, which is also a notable innovation of Spirale. As a result, Spirale can actively adapt to the tortuous external ear canal of different people with lower modulus and smaller contact area under electrothermal actuation, rather than simple expansion and

deformation of an equal-diameter coiled spring. This is also supported by the illustration in the lower-right inset of Fig. 1b in the revised manuscript and the *in vitro* experiments in Supplementary Videos 4 and 5. To vividly depict the morphology of SpiraleE before and after deformation in tubes with nonuniform curvatures, we present the screenshots from the videos in Fig. r1.

In view of your comments, we have deleted the phrase “*via active deformation*” in the title and made appropriate changes to describe the deformation mechanism of SpiraleE at the corresponding position in the modified manuscript.

Fig. r1 | Images of SpiraleE expands and deforms in tubes with nonuniform curvatures.

Modification to the manuscript:

The title has been replaced with “**Conformal in-ear bioelectronics for visual and auditory brain-computer interfaces**”. Additionally, we have replaced the phrase “*via active deformation*” with “**under electrothermal actuation**” at the corresponding place in the modified text, which accurately describes the deformation mechanism of SpiraleE.

In addition, the numbering of figures and videos has also been changed correspondingly.

Comment 5

Line 66: is it really a smaller contact area? Do you compare contact areas to other works?

Our response:

We express our appreciation to the reviewer for this valuable comment. In comparison with other in-ear EEG devices utilizing full coverage, the novel spiral configuration implemented in

SpiralE marks the first application of such technology for in-ear EEG monitoring. After a rough consideration of the cylindrical shape of the ear canal with a constant diameter of 6 mm and the device's penetration depth of 15 mm, the contact area for full coverage is estimated to be approximately 283 mm^2 ($\pi * 6 \text{ mm} * 15 \text{ mm}$). In contrast, the contact area of SpiralE is calculated to be 150 mm^2 ($50 \text{ mm} * 3 \text{ mm}$). Thus, the former boasts a contact area almost twice that of the latter.

This particular design feature boasts a reduced contact area, thereby facilitating diminished damage to the electronics and the ear canal, rendering it more amenable to acceptance by participants. Table r1 (Supplementary Table 1) presents a thorough comparison of the electrode and device contact areas, as well as other features, with previous studies on ear EEG devices. Furthermore, we have included the electrode and device contact areas in the revised manuscript to elucidate the capacity of SpiralE to monitor in-ear EEG with a smaller contact area.

Modification to the manuscript:

On lines 112-116, page 6, we added the following sentences, “In addition, SpiralE has been designed as a long strip with an in-plane size of 50 mm * 3 mm (where the electrode is a circle with a diameter of 1 mm) and is supported on the ear canal with a spiral shape to finish in-ear EEG recording, rather than the previous devices that have full contact with the inner wall of the ear canal (Supplementary Table 1). This strategy further reduces abrasion and friction with the ear canal.”

Comment 6

Line 80: Overall, these results are exciting. From what I could see, other ear eeg works have only been demonstrated with simpler experiments.

Our response:

We extend our sincere gratitude to the reviewer for this valuable comment that has enriched the main text. While previous studies utilizing ear EEG have employed simple experiments to evaluate the feasibility of their devices, they have not employed in-ear bioelectronics as a potent tool for BCIs. In contrast, our visual and auditory BCIs based on SpiralE, notably the online 40-target SSVEP-BCI system (Supplementary Video 6), have successfully integrated in-ear bioelectronics into BCIs, yielding favorable outcomes. To the best of our knowledge, our work is the only study that has accomplished up to 40-target online SSVEP-BCI solely through in-ear EEG

recording. The state-of-the-art SpiraleE in the visual BCI offers an ITR (the most valuable parameter for evaluating BCI performance) of 36.86 ± 15.53 bits/min, the highest value reported thus far in ear EEG research, indicating the feasibility and practicality of utilizing SpiraleE to establish SSVEP-BCI applications. As suggested by the reviewer, we provided a detailed comparison between SpiraleE-based BCI studies and other ear EEG works in Table r1 (Supplementary Table 1).

Modification to the manuscript:

None.

Comment 7

Line 89: Grammar error: "SpiraleE can conform" not conformal

Our response:

We express our gratitude to the reviewer for this valuable observation, which has led us to rectify the grammatical infelicity in our manuscript.

Modification to the manuscript:

On line 91, page 5, we changed the sentence, “SpiraleE can conformal to” to “SpiraleE can conform to”.

Comment 8

Page 5 in general:

Discussions of deformation and thin films should be followed by fatigue or lifetime measurements. Do you foresee this as a single-use, short-term use (a handful of measurements), or long-term use (year+)?

Our response:

We express our gratitude to the reviewer for providing this insightful comment. First, Fig. 2b in the previous manuscript displays the impedance spectra of all five in-ear channels during five repeated expansions and deformations, which demonstrate the electrochemical stability of SpiraleE. In addition, we have supplemented the dynamic resistance change of the weakest parts of SpiraleE, namely, the stretchable electrodes, during 400 cycles of 10% tensile strain in Fig. r2 (Supplementary Fig. 5). The resistance changes during repeated stretching are within an acceptable

range, confirming the good stability and consistency of stretchable parts during repeated deformation.

Based on our tests of Spirale’s fundamental constituents and its holistic evaluation, we are confident that Spirale is fully capable of accomplishing multiple measurements within a time frame (i.e., short-term deployment).

Fig. r2 | The dynamic resistance changes of the stretchable parts of Spirale (including all five in-ear EEG channels and electrothermal actuation layer) under 400 cycles of 10% tensile strain.

Modification to the manuscript:

On lines 150-156, page 8, we added the following sentences, “To validate the fatigue characteristics of Spirale, we investigate the dynamic resistance modulation of its most vulnerable constituents, namely, the stretchable electrodes (comprising the EAL and EEGDL), subjected to 400 cycles of 10% tensile strain, as depicted in Supplementary Fig. 5. Our analysis ascertains the efficacy of Spirale, as evidenced by the consistent resistance modulation observed during the stretching process and negligible alteration in resistance post-stretching. These results corroborate

the feasibility of utilizing SpiraleE in repetitive or short-term applications, such as for a handful of measurements.”

In addition, the numbering of figures has also been changed correspondingly.

Comment 9

Line 103 again alludes to the fact that this does not occlude hearing, but the earlier references also do not occlude hearing. Unfortunately, a hollow assembly is not pushing the state of the art anymore.

Our response:

We express our gratitude to the reviewer for their valuable comment. We acknowledge that some previous studies have also proposed the use of a hollow structure to avoid occlusion of hearing, with Kaveh¹ being a notable example. However, we observed that most of these designs feature a small hole at the end of the device, which we suspect may still partially occlude hearing. Whilst Kaveh’s study is undoubtedly inspiring, we have discovered that the 3D-molded support structure, despite its hollow and user-generic design, lacks adjustability and adaptability to individual canal shapes. We believe that this rigid and solid design may result in reduced user experience. In contrast, we contend that our soft electronics design is better suited to leverage the advantages of hollowness, accommodate user-to-user variation, and enhance user experience, making it a more suitable option for daily use.

Modification to the manuscript:

On lines 102-103, page 8 in the original manuscript, we deleted the sentence, “Compared with the aforementioned ear EEG electronics, this strategy removes the often used supports and allows real-time communication with the outside world in a wearable and discreet manner.”

Reference:

- 1 Kaveh, R. *et al.* Wireless user-generic ear EEG. *IEEE Trans. Biomed. Circuits Syst.* **14**, 727-737, (2020).

Comment 10

Line 146: I request you also report the impedance at 50 Hz (or something that is inside the EEG band of interest).

Our response:

We thank the reviewer for this useful comment. The Spirale-skin impedance spectroscopy of all five in-ear channels has been characterized using the Impedance Analysis Interface (PSM 1700, Newtons4th Ltd), and the results are presented in Figure r3. Specifically, the impedance at 50 Hz for all in-ear channels is found to be 110.96 ± 23.31 kohm, and the information has been included in Table r1 (Supplementary Table 1). The impedance values for all channels at frequencies ranging from 1 to 10^6 Hz can be found in “Source Data”.

Fig. r3 | Spirale-skin impedance spectroscopy of all five in-ear channels.

Modification to the manuscript:

The impedance at 50 Hz has been added to the corresponding location in Supplementary Table 1, “111 kohm @ 50 Hz”.

In addition, the numbering of figures has also been changed correspondingly.

Comment 11

Line 191: you refer to these as active electrodes but I don't see any amplifiers near the electrodes. Are they really active electrodes? Furthermore, there is still no description of "active deformation". It seems like the spiral is just using the built in spring constant from the SMP layers (and the whole assembly I suppose). Furthermore, what are the areas of the electrodes? Impedance of an electrode isn't that helpful without the area. Furthermore, it is important to report the frequency at which you measure impedance (<50kOhms at what frequency?).

Our response:

We are grateful to the reviewer for these insightful comments and apologize for any confusion or ambiguity that may have arisen from the previous manuscript.

Regarding the term “active electrodes” used in our original manuscript, we acknowledge that it may have been misleading. We have revised the description to “available electrodes” in our revised version to better reflect their intended meaning as acceptable or usable electrodes with a resistance of less than 50 kohm at 1 kHz.

Furthermore, we recognize that the phrase “active deformation” is inappropriate, and we have made the necessary corrections to clarify Spirale’s deformation mechanism. Specifically, we have removed this phrase from the title and modified the text to describe Spirale’s adaptive conformity to the inner wall of the external auditory canal after actuation by the electrothermal drive layer. With utmost humility, we earnestly hope that the alterations made have adequately addressed the comments expressed by the reviewer.

In addition, we have added the area of the electrode (a circle with a diameter of 1 mm) and the contact area of the device (50 mm * 3 mm) in the revised manuscript and Table r1 (Supplementary Table 1).

Finally, we acknowledge the omission of the frequency when measuring impedance in the previous manuscript and have supplemented this information in the revised version. An impedance less than 50 kohm is the impedance presented by the Neuroscan system at 1 kHz.

Overall, we deeply regret any confusion or misunderstandings that may have resulted from the previous manuscript. Drawing from the valuable insights of the reviewer, we have made meticulous revisions to address the aforementioned concerns. It is our ardent hope that the revised manuscript will fulfill the reviewer’s expectations, and we extend our gratitude for these valuable comments.

Modification to the manuscript:

On line 213, page 11, we changed the phrase “active electrodes (<50 kohm)” to “available electrodes (<50 kohm @ 1 kHz)”.

The title has been replaced with “Conformal in-ear bioelectronics for visual and auditory brain-computer interfaces”. Additionally, we have replaced the phrase “via active deformation” with “under electrothermal actuation” at the corresponding place in the modified text.

On lines 112-116, page 6, we added the following sentences, “In addition, Spirale has been designed as a long strip with an in-plane size of 50 mm * 3 mm (where the electrode is a circle

with a diameter of 1 mm) and is supported on the ear canal with a spiral shape to finish in-ear EEG recording, rather than the previous devices that have full contact with the inner wall of the ear canal (Supplementary Table 1). This strategy further reduces abrasion and friction with the ear canal.”

Comment 12

Figures 3 & 4 are beautiful. I would suggest making the text slightly larger but that is nitpicky. ***Our response:***

We extend our heartfelt gratitude for your kind acknowledgment of the merits of Figures 3 & 4. In response to your valuable feedback, we have diligently enlarged the font size in the figures and made certain adjustments to enhance its overall aesthetic appeal, as reflected in the revised manuscript.

Modification to the manuscript:

Fig. r4 | Visual BCIs based on Spirale.

Fig. r5 | SpiraleE for auditory BCI.

Summary Comment

Overall, this paper presents a novel in-ear electrode architecture (a spiral vs. traditional earplugs or ear molds) as well as multiple BCI relevant experiments to showcase it's application. This spiral architecture is user-generic but requires hydrogel and skin cleaning (this was hidden in the methods and should be clearly stated in the results - especially when comparing oneself to dry-electrode ear eeg works). Unfortunately, It is unclear how practical this sort of electrode architecture may be in real-world scenarios (hearing aids, day-to-day BCIs, etc.) - how often can it be re-used? Do hydrogels need to be reapplied?... These sorts of questions are left unanswered. While the

discussion claims that this work introduces a new communication channel for overt visual and covert auditory BCIs, I don't quite see if that is a major claim. This work did not invent the experiment or the channel (ear eeg seems to have been around for over a decade). They do, however, demonstrate BCI experiments with greater fidelity and accuracy than I've seen for ear eeg works. Regardless, the main novelty of this work is the electrode technology. As a result, it would be beneficial for a clear discussion/comparison versus other electrode technologies (maybe discuss area vs. impedance or compare to conventional wet electrodes).

Our response:

We would like to express our sincere gratitude to you for your recognition of this work and extend our sincerest apologies for the confusion caused by the unclear and inappropriate description in the original manuscript. To address these issues, we have carefully considered your comments, conducted additional experiments, and provided point-by-point descriptions.

First, it is important to note that SpiralE, as proposed in this work, is based on dry electrodes. Given that dirt and debris can accumulate within the ear, we required participants to clean their earwax with alcohol tips during preparation to achieve acceptable impedance levels. The application of gel has been used only as a last resort for some participants whose in-ear channels could not be reduced to less than 50 kohm @ 1 kHz through other means. Out of the 18 SpiralEs (9 subjects * 2 SpiralEs) involved in the visual experiments, a total of 10 had gel applied. These particular SpiralEs can be identified by their blue highlighting in Table r2 (part of the Supplementary Table 2). To assess the impact of gel application on impedance during epidermal measurements, we measured SpiralE-skin impedance using an Impedance Analysis Interface (PSM 1700, Newtons4th Ltd) and present the data in Fig. r3. In conclusion, while the use of gel offers convenience and enhances the stability of electrophysiological contact, the conformed SpiralE without gel still yields robust EEG data, albeit with an increase in impedance of ~20 kohm.

Subsequently, the application of gel on the electrode is a relatively simple method, whereby a small amount of gel is applied to the electrodes of some participants after SpiralE has been removed from the ear when the impedance is found to be outside the acceptable range (≤ 50 kohm @ 1 kHz). After the experiment, the gel could be easily removed using a cotton swab and water, and participants did not report any discomfort caused by the gel throughout the ~2-hour experiment.

Table r2. The impedance of in-ear channels for all subjects in the visual test (kohm).

	L1	L2	L3	L4	L5	R1	R2	R3	R4	R5	Channels availability
S1	65.7	21.9	76.5	48.9	41.2	42.8	-	36.1	37.2	35.5	7/10
S2	38.2	30.8	44.3	43.2	50.3	76.6	82.2	87.1	73.3	113.1	4/10
S3	44.0	33.5	27.4	36.3	15.7	40.4	39.5	110.0	42.7	25.1	9/10
S4	12.8	9.9	13.0	12.5	13.2	46.8	26.0	17.5	21.4	40.4	10/10
S5	6.2	6.7	9.0	20.8	7.4	3.6	3.6	4.1	4.1	3.9	10/10
S6	12.1	32.6	12.6	24.3	45.9	7.0	6.1	44.6	50.8	70.8	8/10
S7	15.7	20.6	7.6	9.5	49.7	29.1	11	15.2	9.4	65.2	9/10
S8	11.9	5.6	8.6	7.7	<1	41.7	40.9	14.4	17.0	25.8	10/10
S9	15.3	7.6	9.5	4.0	7.32	7.9	14.4	7.4	13.6	7.5	10/10

Whether or not gel is applied, SpiraleE offers reusability and is highly suitable for real-world applications. The electrical stability of SpiraleE is maintained before and after repeated deformation, which is proven by the negligible resistance change of the stretchable electrodes during 400 cycles of 10% tensile strain, as presented in Fig. r2 (Supplementary Fig. 5). The impedance spectra of all five in-ear channels during five repeated expansions and deformations further demonstrate the electrochemical stability of the entire SpiraleE. Additionally, SpiraleE's shape can be repeatedly reset to a small spiral and electrothermally actuated to adapt to various ear canals of different subjects for in-ear EEG monitoring. The successful visual and auditory BCI on 14 subjects further supports the feasibility of SpiraleE's application in real-world scenarios.

Furthermore, we acknowledge the inappropriate description in the original manuscript that suggested that SpiraleE introduced a new communication channel for overt visual and covert auditory BCIs. Rather, our intention was to emphasize the use of SpiraleE in building an online 40-target SSVEP-BCI system and achieving 75.57% classification accuracy (Supplementary Video 6), which represents a significant milestone as the only work that supports up to 40 targets online SSVEP-BCI using ear EEG recording alone (Table r1, also known as Supplementary Table 1). In response to your feedback, we have made necessary adjustments to the revised manuscript.

Moreover, we appreciate your highlighting of the advantages of our electrode technology, which were previously missing from the manuscript. In the revised version, we emphasize the advantages of SpiraleE electrodes, such as their ability to switch modulus for various stages of insertion-deformation-detection-extraction (Fig. r6, also known as Supplementary Fig. 1) and the

smaller contact area that reduces damage to the inner wall of the ear canal. We have also included a detailed comparison of SpiraleE with other ear EEG works in Table r1 (Supplementary Table 1).

Fig. r6 | SpiraleE experiences changes in its storage modulus as it goes through the stages of insertion, deformation, detection, and extraction.

Lastly, this work makes certain claims that are misleading, these electrodes are not "actively deformed", they are deformed and then allowed to expand inside a closed space. This is a coiled spring that they are releasing. Active deformation implies a driving force (like a motor or microfluidics). Secondly, these are not active electrodes. From my understanding, active electrodes require electrode amplifiers right beside the electrode interface in order to minimize the effects of motion and wireline interference.

Our response:

We apologize for any confusion caused by the wording in our original manuscript and thank you for bringing it to our attention. To clarify, the term “active deformation” in our original manuscript refers to SpiraleE’s ability to conform to the shape of the external auditory canal using an electrothermal drive layer to actuate the whole bioelectronics. By integrating an electrothermal actuation layer with SMPs and EEG detection layers, we are able to achieve a unified system for drive, deformation, and EEG monitoring, which represents a significant innovation of SpiraleE. In response to your feedback, we have removed the phrase “via active deformation” from the title and adjusted our description of SpiraleE’s deformation mechanism in the revised manuscript.

Specifically, we highlight that SpiraleE can adapt to the unique shape of each individual’s nonuniform ear canal through electrothermal actuation, resulting in a lower modulus and smaller

contact area rather than a simple coiled spring with equal diameter. This is further supported by the illustration in the lower-right inset of Figure 1b and the results of *in vitro* experiments shown in Supplementary Videos 4 and 5.

In response to your comments, we acknowledge that the use of the term “active electrodes” in our original manuscript may have caused confusion. In our revised version, we have updated the term to “available electrodes” to better convey their intended meaning as electrodes that are acceptable or usable with a resistance of less than 50 kohm at 1 kHz.

All in all, we have made the necessary adjustments to the manuscript to address your concerns and hope that these changes meet with your approval.

Unfortunately, I cannot recommend this for publication in nature due to its smaller scope and lack of practicality. This does not present an entire system, it really presents just an electrode technology (may be a better fit for advanced materials or sensors). That being said, the methodology and figures are all sound/beautiful. It really is a great work. I just don't think it's right for nature.

Our response:

We appreciate your concerns regarding the practicality and limitation of our work. We value the reviewer’s constructive feedback, as it provides additional opportunities for enhancing our work. Although we did not emphasize the integrity of the whole system in our study, the proposed in-ear bioelectronics, when combined with existing commercial amplifiers, constitute a practical BCI system. It is important to note that our approach differs from the regular device validation protocol, which typically only analyzes offline signals under simple experimental paradigms. Instead, we use SpiraleE to build an online 40-target SSVEP-BCI system and achieve 75.57% classification accuracy (Supplementary Video 6), which is the only work that supports up to 40 targets online SSVEP-BCI just relying on ear EEG recording (Supplementary Table 1). We have demonstrated that the SpiraleE functions not only as a mere electrode but also as a mature BCI system that even supports online spelling tasks.

Given the novelty of our in-ear EEG bioelectronics design and the thoroughness of our in-ear BCI applications, we have meticulously revised our manuscript in light of your valuable comments, aspiring to meet the esteemed standards of *Nature Communications*.

Reviewer #2:

Summary Comment

Z. Wang et al. reported an in-ear bioelectronic device, SpiraleE, which can be conformally integrated with auditory meatus. This can be achieved by active thermal deformation of encapsulating shape-memory polymers, enabling facile insertion and adaptive expansion to achieve conformal contact in a spiral along with the inner wall of ear canal. The stretchable EEG electrodes as well as an electrothermal actuator offered mechanical reliability for the in-ear integration. Using the SpiraleE, the authors demonstrated in-ear EEG detection, which can be used in SSVEP BCI classification and 40-target online speller experiments as well. Because these concepts, analysis, and demonstrations are impressive, I recommend this work to be published in Nature Communications after addressing minor issues.

Our response:

With gratitude from the depths of our hearts, we extend our thanks to the reviewer for these positive comments on this work. In accordance with the reviewer's queries, we have diligently addressed each concern and have made the necessary alterations to the revised manuscript.

Comment 1

A major issue of wearable electronics is a bulky external processor for recording and processing EEG data, although the bioelectronic device is soft and thin. These processors are difficult to be integrated with the body, causing discomfort and irritation. Please discuss the integration strategies of in-ear bioelectronics with the external processor.

Our response:

We appreciate the reviewer's useful comments. The integration of the sensing units and the external processor is indeed a challenging and tricky issue, especially in the case of low signal-to-noise ratio in-ear EEG monitoring. In our work, we used a series of flexible anisotropic conductive film, also known as ACF¹⁻⁴, to thermally bond the electrothermally actuated layer (EAL) and the EEG detection layer (EEGDL) onto a personalized PCB board. Furthermore, we joined the PCB board to a DC power and Neuroscan system's amplifier with DuPont wires and alligator clips. With these measures, we established a steadfast link between the EEGDL and amplifier, as well as a reliable connection between the EAL and DC power, as shown in Fig. r1 (Supplementary Fig. 6). Presentation of stimuli and processing of data were done through external equipment. While

we cannot guarantee the flexibility and wearability of the backend processing units, we ensure the flexibility of all parts in contact with skin, which balanced the stability of the interconnection and the comfort of all subjects.

Fig. r1 | The integration between SpiralE and the external processor.

Modification to the manuscript:

On lines 412-418, page 23, we added the following sentences, “It is noteworthy that the fusion of SpiralE with the external processor has been accomplished *via* a sequence of flexible anisotropic conductive film, commonly referred to as ACF, customized printed circuit board (PCB), DuPont wires, and EEG crocodile clip cables. Specifically, the ACF-facilitated thermal bonding process has been employed to secure the EAL and EEGDL connected to the tailored PCB, subsequently allowing for the establishment of interconnects between the EAL and a DC power source, as well as between the EEGDL and an EEG amplifier *via* DuPont wires and alligator clips (Supplementary Fig. 6).”

In addition, the numbering of figures has also been changed correspondingly.

References:

- 1 Viventi, J. *et al.* Flexible, foldable, actively multiplexed, high-density electrode array for mapping brain activity in vivo. *Nat. Neurosci.* **14**, 1599-1605, (2011).
- 2 Norton, J. J. S. *et al.* Soft, curved electrode systems capable of integration on the auricle as a persistent brain-computer interface. *Proc. Natl. Acad. Sci. U. S. A.* **112**, 3920-3925, (2015).
- 3 Tian, L. *et al.* Large-area MRI-compatible epidermal electronic interfaces for prosthetic control and cognitive monitoring. *Nat. Biomed. Eng.* **3**, 194-205, (2019).

- 4 Han, M. *et al.* Catheter-integrated soft multilayer electronic arrays for multiplexed sensing and actuation during cardiac surgery. *Nat. Biomed. Eng.* **4**, 997-1009, (2020).

Comment 2

Another issue is the interconnections between the flexible device and the external processor. The wirings can be easily disconnected, limiting dynamic physical activities. Please describe how the authors interconnected the SpiraleE to the external measurement setup and discuss its reliability for continuous in-ear EEG recording.

Our response:

We appreciate the reviewer's useful comments. The challenge of interconnection has long impeded the deployment of flexible electronics across various applications. In this study, we employed flexible anisotropic conductive film (ACF), PCB, pins, DuPont cables, and EEG alligator clip cables to establish interconnection between the flexible device and external processor, as shown in Fig. r2. Specifically, ACF was employed to connect the EAL and EEGDL to distinct channels of the PCB to establish a reliable interconnection between SpiraleE and PCB. Following this, DuPont wires were inserted into the pins that had been soldered to the corresponding channels of the PCB. By attaching DuPont wires of distinct channels to both a DC power and an EEG amplifier *via* interconnecting cables with alligator clips and terminals compatible with external processors, we achieved a dependable linkage between the EAL and EEGDL of SpiraleE to a DC power and an EEG amplifier, respectively. Although this form of interconnection has limited practicality beyond the laboratory, we can extend the cables length to position the external processor in alternate equipment, such as backpacks.

Fig. r2 | The interconnections between SpiraleE and the external processor.

The reliability of interconnection plays a pivotal role in ensuring stable signal monitoring and transmission. Commercially utilized DuPont wires and EEG crocodile clip cables demonstrate high stability. Moreover, the constancy of the resistance change in an assembly of stretchable electrodes and ACF wires during 400 cycles of 10% tensile strain substantiates the reliability of this interconnection, as shown in Fig. r3 (Supplementary Fig. 5). Additionally, the exciting results of visual and auditory BCIs on 14 subjects provide compelling evidence for the stability and efficacy of SpiralE and interconnections in facilitating continuous in-ear EEG recording.

Fig. r3 | The dynamic resistance changes of the stretchable parts of SpiralE (including all five in-ear EEG channels and electrothermal actuation layer) under 400 cycles of 10% tensile strain.

Modification to the manuscript:

On lines 150-156, page 8, we added the following sentences, “To validate the fatigue characteristics of SpiralE, we investigate the dynamic resistance modulation of its most vulnerable constituents, namely, the stretchable electrodes (comprising the EAL and EEGDL), subjected to

400 cycles of 10% tensile strain, as depicted in Supplementary Fig. 5. Our analysis ascertains the efficacy of SpiraleE, as evidenced by the consistent resistance modulation observed during the stretching process and negligible alteration in resistance post-stretching. These results corroborate the feasibility of utilizing SpiraleE in repetitive or short-term applications, such as for a handful measurements.”

On lines 412-418, page 23, we added the following sentences, “It is noteworthy that the fusion of SpiraleE with the external processor has been accomplished *via* a sequence of flexible anisotropic conductive film, commonly referred to as ACF, customized printed circuit board (PCB), DuPont wires, and EEG crocodile clip cables. Specifically, the ACF-facilitated thermal bonding process has been employed to secure the EAL and EEGDL connected to the tailored PCB, subsequently allowing for the establishment of interconnects between the EAL and a DC power source, as well as between the EEGDL and an EEG amplifier *via* DuPont wires and alligator clips (Supplementary Fig. 6).”

In addition, the numbering of figures has also been changed correspondingly.

Comment 3

In my opinion, SpiraleE may be stiff because it maintained the spiral shape after conformally implanted in the inner wall of the ear canal. However, the wearable device should be soft to avoid irritation caused by dynamic human motion. In this regard, please provide the mechanical properties (e.g., modulus or stiffness) of SpiraleE at each deformation stage to estimate the device's softness.

Our response:

We appreciate the reviewer’s useful comment. The modulus transition of SpiraleE throughout deformation and recording EEG is important for evaluating the contact between SpiraleE and the inner wall of the ear canal. Here, we use the storage modulus for quantitative characterization, which refers to the index of rebound after deformation of the material, indicating the ability of the material to store elastic deformation energy. Fig. r4a (Supplementary Fig. 3b) illustrates the storage modulus-temperature curves of the two kinds of SMPs. It is noteworthy that the storage modulus of SMPs exhibits a drastic reduction of over 100 times as they transit from a rigid state to a soft state. To elaborate, the modulus of SMP1 could decrease from 400 MPa to 300 kPa with the increase in temperature, while SMP2 experiences a shift from 100 MPa to 300 kPa.

Furthermore, we characterize the storage modulus-temperature curve of Spirale during the insertion-deformation-detection-extraction processes to substantiate our design philosophy, as shown in Fig. r4b (Supplementary Fig. 1). The strategy of switchable modulus of Spirale, which can be manipulated *via* electrothermal actuation, ensures stable in-ear EEG monitoring while simultaneously minimizing potential harm to both ear canal and Spirale.

Fig. r4 | Storage modulus-temperature curves of: **a**, Two kinds of SMPs. **b**, Spirale as it goes through the stages of insertion, deformation, detection, and extraction.

Modification to the manuscript:

On lines 107-112, pages 5-6, we have added the following sentences, “Spirale’s modulus switching capability allows for a reduction from 100 MPa to 400 kPa (Supplementary Fig. 1), thereby ensuring adaptation to different processes. On the one hand, the deformation of Spirale in the ear and the removal from the ear are performed at a lower modulus, thereby mitigating the potential adverse effects on the ear canal. On the other hand, the high modulus provides adequate support to conform to the ear canal, thereby reducing the instability of EEG monitoring caused by motion artifacts.”

In addition, the numbering of figures has also been changed correspondingly.

Comment 4

The authors analyzed the impedance of EEG electrode and claimed that all values were within the acceptable range. Then, is the impedance of EEG electrode still sufficiently low even if EEG is measured without a gel lowering the impedance? Because it seems difficult to insert the gel between the electrode and the inner wall of ear canal that are in conformal contact, the impedance should be low even without gel.

Our response:

We express our deep gratitude for the reviewer’s insightful comments, and with utmost humility, we offer our sincere apologies for the ambiguities that may have arisen from the unclear descriptions in the original manuscript.

While it is true that the application of gel can effectively decrease skin-electronics impedance, this approach also poses potential cleaning challenges in ear. Therefore, this method is merely a last resort for some participants whose in-ear channels cannot be reduced to less than 50 kohm @ 1 kHz through other means. In the visual experiments conducted, a total of 18 SpiralEs (9 subjects * 2 SpiralEs) were utilized. Among these, gel was applied to 10 SpiralEs, which can be identified by their distinct blue highlighting in Table r1 (part of the Supplementary Table 2).

Table r1. The impedance of in-ear channels for all subjects in the visual test (kohm).

	L1	L2	L3	L4	L5	R1	R2	R3	R4	R5	Channels availability
S1	65.7	21.9	76.5	48.9	41.2	42.8	-	36.1	37.2	35.5	7/10
S2	38.2	30.8	44.3	43.2	50.3	76.6	82.2	87.1	73.3	113.1	4/10
S3	44.0	33.5	27.4	36.3	15.7	40.4	39.5	110.0	42.7	25.1	9/10
S4	12.8	9.9	13.0	12.5	13.2	46.8	26.0	17.5	21.4	40.4	10/10
S5	6.2	6.7	9.0	20.8	7.4	3.6	3.6	4.1	4.1	3.9	10/10
S6	12.1	32.6	12.6	24.3	45.9	7.0	6.1	44.6	50.8	70.8	8/10
S7	15.7	20.6	7.6	9.5	49.7	29.1	11	15.2	9.4	65.2	9/10
S8	11.9	5.6	8.6	7.7	<1	41.7	40.9	14.4	17.0	25.8	10/10
S9	15.3	7.6	9.5	4.0	7.32	7.9	14.4	7.4	13.6	7.5	10/10

To assess the influence of gel application on impedance during epidermal measurements, we measured the Spirale-skin impedance using an Impedance Analysis Interface (PSM 1700, Newtons4th Ltd) and present the data in Fig. r5 (Supplementary Fig. 8).

Fig. r5 | Spirale-skin impedance spectroscopy of all five in-ear channels with and without gel.

Furthermore, we have found that achieving acceptable impedance levels relies on cleaning the ear canal of oil and dirt. To this end, each participant cleaned their earwax with alcohol tips during preparation. Then, a small amount of gel is applied to the electrodes for some participants after Spirale has been removed from the ear when the impedance is found to be outside the acceptable range (≤ 50 kohm @ 1 kHz), rather than inserting the gel after conformal contact. Throughout the ~2-hour experiment, participants did not report any discomfort caused by the gel. The gel was removed using a cotton swab and water after the experiment.

In brief, while the gel offers convenience and enhances the stability of electrophysiological contact, the conformed Spirale without gel still yields robust EEG data, albeit with an increase in impedance of ~20 kohm.

Modification to the manuscript:

On lines 204-208, page 11, we have added the following sentences, “Supplementary Fig. 8 displays the Spirale-skin impedance of in-ear channels, with and without gel, as determined via an Impedance Analysis Interface (PSM 1700, Newtons4th Ltd). Despite the convenience and enhanced stability of electrophysiological contact provided by the gel, our experiments demonstrate that the conformed Spirale without gel still produces robust EEG data, with an impedance increase of approximately 20 kohm at 1000 Hz.”

On lines 440-451, pages 24-25, we have changed the sentences to: “During this process, an ear endoscope (Bebrid Note3) was used to inspect the deformation and assisted Spirale to fit snugly into the ear canal. In case the impedance value still cannot be lowered into the acceptable

range (≤ 50 kohm @ 1 kHz), the SpiraleE was taken out and reshaped into a small spiral. A small amount of gel was applied to SpiraleE, and it is then reinserted into the ear canal and actuated again. Out of the 18 SpiraleEs (9 subjects * 2 SpiraleEs) involved in the visual experiments, a total of 10 had gel applied. These particular SpiraleEs can be identified by their blue highlighting in Supplementary Table 2. When the power was removed, SpiraleE was self-supported in ear canal after cooling to room temperature. Similarly, a voltage was applied to SpiraleE after the end of the experiment. After the modulus decreased, SpiraleE was carefully removed and cleaned using cotton swabs and water for reuse by next subject. The subjects were also instructed to clean their external auditory meatus using cotton swabs and alcohol to ensure proper hygiene. Notably, all participants did not report any discomfort throughout the ~2 hours experiment, irrespective of the presence or absence of gel.”

In addition, the numbering of figures has also been changed correspondingly.

Comment 5

It would be helpful to provide detailed information on SMP1 and SMP2 (e.g., molecular structure and polymer network) for better understanding.

Our response:

We appreciate the reviewer’s useful comments. The molecular structure and polymer network of the precursor monomers and the synthesized SMPs involved have been added to Fig. r6 (Supplementary Fig. 2). SMP1 and SMP2 have similar structures, but the molecular weights of polycaprolactone diol (PCL) in SMP1 and SMP2 are different. The molecular weight of PCL in SMP1 is 2000 g/mol, while it is 10000 g/mol in SMP2.

Fig. r6 | The molecular structure and polymer network of the precursor monomers and the synthesized SMPs. **a**, PCL. **b**, PHMD. **c**, HDI. **d**, The synthesized SMPs.

Modification to the manuscript:

On lines 368-372, page 21, we added the following sentences, “The molecular structure and polymer network of the two kinds of SMPs involved are shown in Supplementary Fig. 2. SMP1 and SMP2 have similar structures, but the molecular weights of polycaprolactone diol (PCL) in SMP1 and SMP2 are different. The molecular weight of PCL in SMP1 is 2000 g/mol, while it is 10000 g/mol in SMP2.”

In addition, the numbering of figures has also been changed correspondingly.

Comment 6

The authors claimed that ‘ESCL is the SMP2 layer’. What is ESCL?

Our response:

We appreciate the reviewer’s useful comments. There was an improper abbreviation or typo on line 114: ‘Next to ESCL is the SMP2 layer’. As a result, ‘ESCL’ should be changed to ‘EEGDL’.

Modification to the manuscript:

On line 123, page 6, we changed the sentence to: “Next to EEGDL is the SMP2 layer,”

Reviewer #3

Summary Comment

The authors introduce a conformal in-ear EEG sensor named SpiraleE for visual and auditory brain-computer interface (BCI). Joule heating can expand the spiral-shaped electrodes supported by a shape memory polymer (SMP), conforming to various ear canal shapes. Because the electrode does not require rigid supports, such as earphones or ear plugs, the user's auditory perception is unobstructed. The reported device has novelty, and the authors successfully demonstrated its potential as BCI. Videos and supplementary materials are very helpful for understanding the work. The authors may consider the following points to further improve the manuscript.

Our response:

We express our gratitude to the reviewer for the positive comments. With great attentiveness and conscientiousness, we have addressed the raised concerns and implemented essential modifications to our manuscript. We earnestly hope that our responses fulfill your expectations and requirements.

Comment 1

The mechanical properties of SpiraleE, such as its thickness and modulus, should be explicitly reported to support the ear conformability of the electronics. Also, the ear canal is not perfectly cylindrical, and there can be localized curvatures inside. Please comment on whether the SpiraleE is soft enough to conform to nonuniform curvatures.

Our response:

We express our gratitude for the valuable comments provided by the esteemed reviewer. The original manuscript presented the storage modulus-temperature curves for the two kinds of SMPs in Fig. r1a (Supplementary Fig. 3b). It is noteworthy that the storage modulus of SMPs undergoes a remarkable reduction of over 100 times as they transition from a rigid to a soft state. To illustrate, the modulus of SMP1 could decrease from 400 MPa to 300 kPa with a rise in temperature, while SMP2 experiences a shift from 100 MPa to 300 kPa. Additionally, we characterize the storage modulus-temperature curves of SpiraleE during the insertion-deformation-detection-extraction processes (Fig. r1b) in the revised manuscript (Supplementary Fig. 1) to substantiate our design strategies. By electrothermally actuating SpiraleE, we can readily switch the modulus, promoting in-ear EEG monitoring stability and mitigating ear canal abrasions.

Fig. r1 | Storage modulus-temperature curves of: **a**, Two kinds of SMPs. **b**, SpiraleE as it goes through the stages of insertion, deformation, detection, and extraction.

Furthermore, the external auditory canal is, in truth, a winding semi-closed cavity with nonuniform curvature, placing exceedingly high demands on the stiffness and deformation capacity of the in-ear EEG device. SpiraleE conforms adaptively to the inner wall of ear canal with a sufficiently low modulus (400 kPa) during deformation. We have provided additional evidence in the form of Supplementary Video 5 to show that SpiraleE is capable of adaptively fitting inside complex tubes with varying curvatures after electrothermal actuation. This, in conjunction with the *in vitro* experiments presented in Fig. 2e and Supplementary Video 4 in the original manuscript, provides vivid verification and explanation of SpiraleE’s ability to adaptively conform to nonuniform curvatures. To vividly depict the morphology of SpiraleE before and after deformation in tubes with nonuniform curvatures, we present the screenshots from the videos in Fig. r2. Moreover, the irregular three-dimensional structure of SpiraleE with a variable radius along the axial direction (lower-right inset of Fig. 1b) after being extracted from the ear canal also serves as a demonstration of this capability.

Fig. r2 | Images of SpiraleE expands and deforms in tubes with nonuniform curvatures.

Modification to the manuscript:

On lines 107-112, pages 5-6, we have added the following sentences, “SpiraleE’s modulus switching capability allows for a reduction from 100 MPa to 400 kPa (Supplementary Fig. 1), thereby ensuring adaptation to different processes. On the one hand, the deformation of SpiraleE in the ear and the removal from the ear are performed at a lower modulus, thereby mitigating the potential adverse effects on the ear canal. On the other hand, the high modulus provides adequate support to conform to the ear canal, thereby reducing the instability of EEG monitoring caused by motion artifacts.”

On lines 183-185, pages 9-10, we have changed the sentence to: “It can expand and adaptively conform to the inner wall of the pipe, showing the novel ability of SpiraleE to fit complex nonuniform curvatures under electrothermal actuation, as illustrated in Fig. 2e, Supplementary Videos 4 and 5.”

In addition, the numbering of figures and videos has also been changed correspondingly.

Comment 2

Fig. 2a and 2b show the impedance spectroscopy of the device itself, not an electrode-skin contact impedance. Although the figure illustrates that the impedance (primarily resistance in this case) is not affected by expansion, it cannot fully verify the stability of the device as bioelectronics. Supplementary Fig. 3 shows the electrode-skin impedance recorded by an EEG device, but biogel

was applied for this measurement, diminishing its advantages as a dry thin-film-based electrode. The authors should perform the electrode-skin contact impedance spectroscopy without gel to make a fair comparison with conventional EEG systems and other dry electrodes in the literature. **Our response:**

We are grateful for the reviewer’s valuable comments. We have included the Spirale-skin impedance data, with and without gel coating, as measured *via* an Impedance Analysis Interface (PSM 1700, Newtons4th Ltd), in the revised manuscript and depicted it in Fig. r3 (Supplementary Fig. 8). As expected, the application of gel coating results in a considerable reduction in skin-electronics impedance by an order of magnitude. With regards to convenience of use, we recommend using gel coating only as a last resort for those participants for whom in-ear channel impedance cannot be reduced to less than 50 kohm @ 1 kHz using alternative approaches.

Fig. r3 | Spirale-skin impedance spectroscopy of all five in-ear channels with and without gel.

In accordance with the reviewer’s comments, we have conducted a quantitative comparison of the impedance of Spirale-skin in the absence of gel with other ear EEG devices. This comprehensive analysis is presented in Table r1 (Supplementary Table 1), combined with other characteristics. Obviously, Spirale, equipped with dry thin-film electrodes, remains a highly effective device for in-ear EEG monitoring.

Table r1. Comparison of recent ear EEG electronics.

		Ret¹	Ret²	Ret³	Ret⁴	Ret⁵	Ret⁶	Ret⁷	Ret⁸	Ret⁹	This study
Electrodes Properties	Location	In-ear	In-ear	Auricle, mastoid	In-ear	Behind ear	Auricle, in-ear	In-ear	Auricle, in-ear	Auricle, in-ear	In-ear
	Support	Earplug	Earphone	-	Earplug	Cap	Earmould silicone	Earmould silicone	Resin (PC/PI)	Resin (PC/PI)	SMPs
	Contact Area	Whole canal	Whole canal	-	Whole canal	Full head	Whole canal	Whole canal	Whole canal	Whole canal	Spiral
	Electrode Area	-	-	-	-	-	9.6 mm ²	-	60 mm ²	-	0.785 mm ²
	Electrode Material	Ag	CNT/PDMS	Au	Wet Ag coated nylon	Ag/AgCl	Ti/IrO ₂	IrO ₂	Ag	Ag	Au
	Skin Impedance (50 Hz)	5 k	~150 k	-	10 k	-	435 k	-	392 k	190 k	111 k
	Sensors	8	1	5	2	8	8	12	6	10	10
Features	Style	Generic	Generic	Generic	Generic	Generic	Customized	Customized	Generic	Generic	Generic
	Communication	No	No	-	No	-	No	No	Yes	Yes	Yes
	Structure	Stuffed	Stuffed	-	Stuffed	-	Stuffed	Stuffed	Hollow	Hollow	Hollow
Experiments				2-target SSVEP							9-target SSVEP
	Visual	SSVEP	SSVEP	36-target P300: 12 bpm	-	12-target SSVEP: 30.21 ± 10.61 bpm	SSVEP	-	-	-	40-target speller: 36.86 ± 15.53 bpm
	Auditory	ASSR	AEP/ASSR	-	ASSR	-	ASSR	ASSR	ASSR	-	AAD: Accuracy : 72.42%
	Other	Alpha rhythm	Alpha rhythm	Alpha rhythm	-	-	Alpha rhythm	-	Alpha rhythm	Drowsiness detection	Alpha rhythm
	Subjects	14	6	3	5	16	12	10	3	5	14

Modification to the manuscript:

On lines 204-208, page 11, we have added the following sentences, “Supplementary Fig. 8 displays the Spirale-skin impedance of in-ear channels, with and without gel, as determined via an Impedance Analysis Interface (PSM 1700, Newtons4th Ltd). Despite the convenience and enhanced stability of electrophysiological contact provided by the gel, our experiments demonstrate that the conformed Spirale without gel still produces robust EEG data, with an impedance increase of approximately 20 kohm at 1000 Hz.”

In addition, the numbering of figures has also been changed correspondingly.

References:

- 1 Kidmose, P., Looney, D., Ungstrup, M., Rank, M. L. & Mandic, D. P. A study of evoked potentials from ear-EEG. *IEEE Trans. Biomed. Eng.* **60**, 2824-2830, (2013).
- 2 Hoon Lee, J. *et al.* CNT/PDMS-based canal-typed ear electrodes for inconspicuous EEG recording. *J. Neural Eng.* **11**, 046014, (2014).
- 3 Norton, J. J. S. *et al.* Soft, curved electrode systems capable of integration on the auricle as a persistent brain-computer interface. *Proc. Natl. Acad. Sci. U. S. A.* **112**, 3920-3925, (2015).
- 4 Goverdovsky, V., Looney, D., Kidmose, P. & Mandic, D. P. In-Ear EEG from viscoelastic generic earpieces: robust and unobtrusive 24/7 monitoring. *IEEE Sens. J.* **16**, 271-277, (2016).
- 5 Wang, Y. T. *et al.* An online brain-computer interface based on SSVEPs measured from non-hair-bearing areas. *IEEE Trans. Neural Syst. Rehabil. Eng.* **25**, 14-21, (2017).
- 6 Kappel, S. L., Rank, M. L., Toft, H. O., Andersen, M. & Kidmose, P. Dry-contact electrode ear-EEG. *IEEE Trans. Biomed. Eng.* **66**, 150-158, (2019).
- 7 Bertelsen, A. R. *et al.* Generic dry-contact ear-EEG. // 2019 41st Annual International Conference of the IEEE Engineering in Medicine and Biology Society (EMBC), 5552-5555, (2019).
- 8 Kaveh, R. *et al.* Wireless user-generic ear EEG. *IEEE Trans. Biomed. Circuits Syst.* **14**, 727-737, (2020).
- 9 Schwendeman, C., Kaveh, R. & Muller, R. Drowsiness detection with wireless, user-generic, dry electrode ear EEG. // 2022 44th Annual International Conference of the IEEE

Comment 3

Since the clear in-ear montage is not provided, there is no clear justification for having five in-ear channels. As a good BCI carefully selects channels near the source of the EEG modulation to a specific BCI task, authors should provide the proper neuroscientific reasoning, if any, behind the design and channel selection of SpiraleE.

Our response:

We appreciate the valuable feedback provided by the reviewer. The selection of an appropriate EEG acquisition montage is essential for the development of a reliable BCI system. In our proposed SpiraleE-BCI system, we design the placement and number of sensors based on the theoretical spatial limits of noninvasive electrophysiology. Research has shown that EEG has a spatial resolution at the centimeter-level, which indicates that adjacent sensors convey additional information with a minimum distance of one centimeter¹. In addition, data from visual experiments have confirmed that “inter-electrode spacing smaller than 2 cm provides additional neural information”². From both theoretical and empirical evidence, as the total length of the SpiraleE-BCI system is 50 mm, we position the maximum number of electrodes that the device permits.

While we focus on demonstrating the decoding of classical steady-state visually evoked potentials (SSVEPs) and auditory attention decoding (AAD), the proposed ear EEG-BCI can be applied to a broader range of paradigms. However, we acknowledge that the selection of channels should be discussed in the context of specific paradigms. For instance, SSVEPs, which originate from the occipital lobe, can activate a wide range of cortices, suggesting that decoding with a minimal single-channel montage may be possible. However, as a platform, the acquisition module should always match the upper limits of wide range of paradigms, less electrodes can be insufficient for more refined activation coverage.

Furthermore, other works have also chosen to integrate ~10 electrodes into their device³. In this sense, we agree with these studies that that the number of electrodes ought to approach the spatial limits of EEG to provide versatility for various paradigms.

Modification to the manuscript:

On lines 356-360, page 20, we have added the following sentences, “Based on the spatial

resolution of non-invasive neural signals^{1,2}, the length of SpiraleE, and the paradigms used in BCIs, we have selected five in-ear channels for this study. However, future investigations should consider a more in-depth examination of the neurophysiological significance of channels selection in ear, with the aim of achieving full coverage of in-ear EEG information while minimizing costs.”

References:

- 1 Gerven, M. v. *et al.* The brain-computer interface cycle. *J. Neural Eng.* **6**, 041001, (2009).
- 2 Robinson, A. K. *et al.* Very high density EEG elucidates spatiotemporal aspects of early visual processing. *Sci. Rep.* **7**, 16248, (2017).
- 3 Kaveh, R. *et al.* Wireless user-generic ear EEG. *IEEE Trans. Biomed. Circuits Syst.* **14**, 727-737, (2020).

Comment 4

The authors use signal-to-noise ratio (SNR) as a validation metric, but no clear definitions of “signal” and “noise” are provided. As there is no generic definition of signals and noises for EEG, the authors should provide their own definitions and reasoning behind these definitions.

Our response:

We appreciate the insightful comment provided by the reviewer and extend our sincerest apologies for not providing a clear definition of SNR. Indeed, the signal and noise in EEG can be hard to define, but in visual evoked paradigms, especially SSVEPs, the signal component is considered to be the visual evoked part, and the remaining component is considered noise. Here we compute the narrowband SNR where only the neighboring few K frequencies around the stimulation frequency are considered noise¹:

$$SNR_K = \frac{K \cdot |F(f)|^2}{\sum_{k=1}^K |A(f+k)|^2}$$

where f is the stimulation frequency and K denotes the number of neighboring frequencies. The full code is adapted from the MNE tutorial.

Modification to the manuscript:

On lines 496-500, page 27, we have added the following sentences, “We first calculate the narrow band SNR spectrum for 9-target SSVEP¹. Here the noise is considered to be the power

SNR_K = $\frac{\text{spectra density around neighboring } K \text{ frequencies on both sides.}}{K \times F(f)}$

12

$$\frac{\sum_{k=1}^K (F(f) + A(f-k))}{K \times F(f)}$$

where f is the stimulation frequency and K denotes the number of neighboring frequencies ($K = 4$ in this study).”

Reference:

1 Meigen, T., & Bach, M. On the statistical significance of electrophysiological steady-state responses. *Doc. Ophthalmol.* **98**, 207-232, (1999).

Comment 5

Fig. 4e shows that the Pearson’s-correlation-based temporal response function (TRF) models may not offer the best classification since many dots are placed across the boundary. The authors may perform feature analysis or try other machine-learning approaches to further improve the classification.

Our response:

Thank you for your insightful comment. It is true that utilizing more complex machine learning algorithms, such as deep neural networks and spatial filters, can potentially improve high-performance decoding^{1,2}. However, in our study, our primary focus is to demonstrate the potential of Spirale as a novel in-ear EEG acquisition device. Thus, we use basic decoding methods instead of more complicated ones. Additionally, we have found that the TRF reconstruction method, although simple and short in terms of high prediction performance, provides a clearer understanding of the underlying mechanisms involved in decoding auditory stimuli. Nonetheless, we appreciate your suggestion and will consider exploring more advanced machine learning algorithms in future studies.

Despite the introduction of more sophisticated classification methods, it is important to acknowledge that the general performance of decoding natural stimuli remains relatively low. In comparison to “unnatural” paradigms such as auditory oddball, the classification performance of natural stimuli can be substantially lower, i.e., many dots are placed across the boundary³. It is possible that a compromise exists between employing natural or “unnatural” stimuli and achieving low or high separability. Future research may explore more advanced machine learning approaches

or other artificial intelligence methods to enhance the accuracy rate of classification. Alternatively, the present level of classification accuracy may be constrained by the neurophysiological significance of natural speech stimuli. Nonetheless, the reported accuracy of ~80% achieved in our study is comparable to that of other ear EEG investigations³. From the authors' perspective, this level of accuracy underscores the potential of Spirale to confer significant benefits in the field of auditory BCI.

Modification to the manuscript:

On lines 320-324, pages 17-18, we have added the following sentences, “Utilizing Pearson’s-correlation based algorithm, this study achieves an accuracy rate comparable to the current literature reports for auditory BCI classification using in-ear EEG. It is noteworthy that the potential application of more sophisticated machine learning algorithms, including the introduction of artificial intelligence, could potentially enhance the classification accuracy of auditory BCI in the future.”

References:

- 1 Tian, Y., & Ma, L. Auditory attention tracking states in a cocktail party environment can be decoded by deep convolutional neural networks. *J. Neural Eng.* **17**, 036013, (2020).
- 2 Geirnaert, S. *et al.* Electroencephalography-based auditory attention decoding: Toward neurosteered hearing devices. *IEEE Signal Process. Mag.* **38**, 89-102, (2021).
- 3 Fiedler, L. *et al.* Single-channel in-ear-EEG detects the focus of auditory attention to concurrent tone streams and mixed speech. *J. Neural Eng.* **14**, 036020, (2017).

REVIEWER COMMENTS

Reviewer #1 (Remarks to the Author):

Thank you for your updates and additions.

Firstly thank you for the clarification on what you meant by active deformation. Referring to the process as electrothermal deformation is significantly more precise and clear.

Table r1:

References 6, 7, 8 & 9 all have hollow earpieces that allow for communication. The authors claim that the hole is too small in reference 7's earpiece, but they perform auditory experiments so clearly it is large enough to hear through.

Line 162: Do you mean that your system is sampling at 1 kHz or that was the test frequency? It is important for the author to know that regardless if the system is sampling at 1 kHz, the EEG signals at lower frequencies (10-50 Hz) are attenuated/interfered with according to the impedance at (for the sake of this example) 10-50 Hz. That's why reporting impedance at 1 kHz is not really useful.

You make a reference to the electrodes reaching almost 50 degrees Celsius when placed inside the ear canal. While you say it is okay for short periods of time, it's unclear what 'short' means. S7 shows that the electrodes are indeed 49.3C inside the ear. The ear canal looks like it is close to 40C. It only takes a few minutes at that temperature to cause serious burns. Did your IRB allow you to do this?

Reviewer #2 (Remarks to the Author):

The authors have made sufficient revisions to improve the quality of this manuscript. This innovative work can meet the broad readership of Nature Communications, thus I would recommend the publication of this manuscript.

Reviewer #3 (Remarks to the Author):

The authors have made a significant effort to address my comments. The proposed electrothermal activation and adjustability of shape and modulus in the ear are novel and robust. The authors have conducted a fair literature search to highlight the most significant advantages of SpiralE as an occlusion-free in-ear EEG device. I agree with the authors' response that devices like SpiralE can aid in finding in-ear channels with high SNR for future applications. The authors' definition of SNR (narrowband SNR) is rigorous and appropriate for SSVEP quality assessment. Overall, I believe that the revised manuscript presents exciting advancements and would be suitable for publication in Nature Communications. Here are some final suggestions:

1. In regards to the low modulus of 400 kPa and reliable resistance of electrodes under cyclic loading, it is evident that SpiralE is a mechanically reliable flexible electrode. However, it would be beneficial if the authors could provide a reference or calculation to support their claim that 400 kPa is soft enough for good conformability to the ear canal, for instance, by providing information on the representative curvature and Young's modulus of the human ear canal.
2. Regarding the skin contact impedance of 111 kohm, it seems acceptable when compared to recent ear EEG electrodes. Nonetheless, an impedance of less than 10 kohm at 50-60 Hz is recommended for standard EEG, especially for a passive electrode. To put this into perspective, several thin-film-based forehead EEG electrodes have achieved such low impedance with good electrode-skin conformability. This leads me to question whether full skin conformation in the ear canal is still unattainable with SpiralE. It would be useful for the authors to address this limitation and provide insights for further improvement in the manuscript.
3. While I agree that the primary focus of the manuscript is to demonstrate SpiralE's potential as a valid BCI electrode and not the machine learning model, it would be helpful if the authors could provide any differences between the channels or any significance in specific channels for decoding. Such findings could serve as a guideline for future in-ear BCI development.

Responses to reviewers' comments

Reviewer #1

Summary Comment

Thank you for your updates and additions.

Firstly thank you for the clarification on what you meant by active deformation. Referring to the process as electrothermal deformation is significantly more precise and clear.

Our response:

We express our sincere gratitude to you for your recognition of the revised manuscript.

Comment 1

Table r1:

References 6, 7, 8 & 9 all have hollow earpieces that allow for communication. The authors claim that the hole is too small in reference 7's earpiece, but they perform auditory experiments so clearly it is large enough to hear through.

Our response:

We appreciate the comment provided by the reviewer and concur with the assertion that previous studies have utilized a small hole for conducting auditory experiments. Regarding the devices alluded to in references 6, 7, 8, and 9, they are hollow, with the exception of reference 6, which fills the ear canal except for a small hole. Reference 7 designs a hollow earpiece, as stated in the text "A cylindrical conduit was added to the ear canal part to prevent acoustic occlusion of the ear and allow passage of electrode cables." We have rectified this in Table r1 (Supplementary Table 1). It is worth noting, however, that while numerous studies have conducted auditory experiments such as ASSR, they have typically employed headsets or earphones for these traditional, mechanical, and laboratory-level paradigms. These auditory experiments differ significantly from real-world, continuous natural speech stimuli.

In contrast to this approach, our study employs a methodology that accurately simulates ambient sounds by utilizing microphones, thus closely approximating the cocktail effect in real-world settings. If other studies can also receive sounds from the environment, we believe that these studies are also unlikely to significantly impede hearing. Consequently, we contend that the Spirale morphology advanced in this study confers distinct benefits in facilitating communication between individuals and their external surroundings.

We hope this explanation has addressed any confusion, and we extend our gratitude to the reviewer for the valuable comment.

Modification to the manuscript:

In Supplementary Table 1, we have meticulously categorized the characteristics of recent ear EEG electronics, including whether the device is hollow or stuffed, the quantity of channels in a single ear, etc.

Table r1. Comparison of recent ear EEG electronics.

	Ref ¹	Ref ²	Ref ³	Ref ⁴	Ref ⁵	Ref ⁶	Ref ⁷	Ref ⁸	Ref ⁹	This study	
Electrodes Properties	Location	In-ear	In-ear	Auricle, mastoid	-ear	Behind ear	Auricle, in-ear Auricle, in-In	Auricle, in-ear	Auricle, in-ear	ear	In-ear
	Support	Earplug	Earphone	-	Earplug	Cap	Earmould silicone	Earmould silicone	Resin (PC/PI)	Resin (PC/PI)	SMPs
	Contact Area	Whole canal	Whole canal	-	Whole canal	Full head	Whole canal	Whole canal	Whole canal	Whole canal	Spiral
	Electrode Area	20 mm ²	-	-	40 mm ²	-	-	5.3 mm ²	60 mm ²	60 mm ²	0.785 mm ²
	Electrode Material	Ag	CNT/PDMS	Au	Wet Ag coated nylon	Ag/AgCl	Ti/IrO ₂	IrO ₂	Ag	Ag	Au
	Skin Impedance (50 Hz)	-	~150 k	-	10 k	-	435 k	-	392 k	190 k	111 k
	In-ear sensors (each ear)	2	1	-	2	-	3	2	4	4	5
Features	Style	Generic	Generic	Generic	Generic	-	Customized	Customized	Generic	Generic	Generic
	Communication	Yes	No	Yes	No	-	No	Yes	Yes	Yes	Yes
	Structure	Hollow	Hollow	-	Stuffed	-	Stuffed (with a small hole)	Hollow	Hollow	Hollow	Hollow
Experiments	Visual	SSVEP/VEP	SSVEP	2-target SSVEP 36-target P300: 12 bpm	-	12-target SSVEP: 30.21 ± 10.61 bpm	SSVEP	-	-	-	9-target SSVEP 40-target speller: 36.86 ± 15.53 bpm AAD: Accuracy : 72.42%
	Auditory	ASSR/AEP	ASSR/AEP	-	ASSR	-	ASSR	ASSR	ASSR	-	Alpha rhythm
	Other	-	Alpha rhythm	Alpha rhythm	-	-	Alpha rhythm	-	Alpha rhythm	Drowsiness detection	Alpha rhythm
	Subjects	14	6	3	5	16	12	10	3	5	14

References:

- 1 Kidmose, P., Looney, D., Ungstrup, M., Rank, M. L. & Mandic, D. P. A study of evoked potentials from ear-EEG. *IEEE Trans. Biomed. Eng.* **60**, 2824-2830, (2013).
- 2 Hoon Lee, J. *et al.* CNT/PDMS-based canal-typed ear electrodes for inconspicuous EEG recording. *J. Neural Eng.* **11**, 046014, (2014).
- 3 Norton, J. J. S. *et al.* Soft, curved electrode systems capable of integration on the auricle as a persistent brain-computer interface. *Proc. Natl. Acad. Sci. U. S. A.* **112**, 3920-3925, (2015).
- 4 Goverdovsky, V., Looney, D., Kidmose, P. & Mandic, D. P. In-Ear EEG from viscoelastic generic earpieces: robust and unobtrusive 24/7 monitoring. *IEEE Sens. J.* **16**, 271-277, (2016).
- 5 Wang, Y. T. *et al.* An online brain-computer interface based on SSVEPs measured from non-hair-bearing areas. *IEEE Trans. Neural Syst. Rehabil. Eng.* **25**, 14-21, (2017).
- 6 Kappel, S. L., Rank, M. L., Toft, H. O., Andersen, M. & Kidmose, P. Dry-contact electrode ear-EEG. *IEEE Trans. Biomed. Eng.* **66**, 150-158, (2019).
- 7 Bertelsen, A. R. *et al.* Generic dry-contact ear-EEG. // *2019 41st Annual International Conference of the IEEE Engineering in Medicine and Biology Society (EMBC)*, 5552-5555, (2019).
- 8 Kaveh, R. *et al.* Wireless user-generic ear EEG. *IEEE Trans. Biomed. Circuits Syst.* **14**, 727-737, (2020).
- 9 Schwendeman, C., Kaveh, R. & Muller, R. Drowsiness detection with wireless, user-generic, dry electrode ear EEG. // *2022 44th Annual International Conference of the IEEE Engineering in Medicine & Biology Society (EMBC)*, 9-12, (2022).

Comment 2

Line 162: Do you mean that your system is sampling at 1 kHz or that was the test frequency? It is important for the author to know that regardless if the system is sampling at 1 kHz, the EEG signals at lower frequencies (10-50 Hz) are attenuated/interfered with according to the impedance at (for the sake of this example) 10-50 Hz. That's why reporting impedance at 1 kHz is not really useful. **Our response:**

We extend our appreciation to the reviewer for the valuable comments, and also sincerely apologize for any confusion that may have arisen from our previous statements.

In the previous manuscript, we provided a somewhat vague description of the test frequency and sampling frequency. Based on the comments of the reviewer, we have meticulously revised and thoroughly scrutinized the manuscript.

Notably, we emphasize that test frequency and sampling frequency are distinct concepts and must be treated as such. It is essential to acknowledge that impedance and sampling frequency are independent of each other. For EEG measurements, achieving high temporal resolution is critical and often measured at the millisecond level. Hence, to ensure optimal temporal resolution, the sampling frequency of EEG amplifiers is conventionally set at 1 kHz.

It should be noted that our previous statement was not intended to suggest that impedance level was tested solely under 1 kHz. Specifically, we provided impedance values over a frequency range of 10^{-1} - 10^4 Hz and 10^0 - 10^6 Hz, with respect to Electrical Impedance Spectroscopy (EIS) and skin contact impedance spectroscopy tests, respectively. These test frequencies are clearly shown in Fig. r1 (Figs. 2a, 2b, and Supplementary Fig. 11). The impedance magnitude is comparable to those reported in prior researches, as shown in Table r1 (Supplementary Table 1).

The fundamental principle of EIS is based on the generation of a current response signal that corresponds to an AC signal of a sinusoidal voltage applied to the electrode system, from which the electrode impedance can be obtained. The impedance spectrum, produced from a series of frequency sine wave signals, is commonly referred to as EIS. Similarly, skin contact impedance spectroscopy (SpiralE-skin impedance spectroscopy) is also based on a comparable principle. The test frequencies referred to in the preceding rationale are those specified by the reviewer.

Fig. r1 | The impedance spectroscopies. a, EIS test results of the five channels in four states (the shading represents the standard deviation). **b,** EIS test results of the five channels during five

repeated expansions and deformations (the shading represents the standard deviation). **c**,
 Spirale-skin impedance spectroscopy of all five in-ear channels with and without gel.

Additionally, we presented the actual impedance values obtained from the amplifier interface during the BCIs tests (Fig. r2 and Table r2, also referred to as Supplementary Fig. 12 and Supplementary Table 2). Unfortunately, we were unable to determine the corresponding testing frequency for the reported impedance values, despite our efforts to contact NeuroScan Inc., which did not yield a response. We hypothesize that this information may be classified as a commercial secret and, therefore, not readily available. Nonetheless, given that the equipment used was a widely recognized, stable, and standard amplifier, and the impedance values displayed on its interface were recorded at a specific test frequency, the results are comparable with other literatures^{1,2}. Furthermore, we contend that obtaining reliable EEG signals is achievable, as long as the reported impedance values by the commercial amplifier fall within an acceptable range (≤ 50 kilohms). This claim is supported by the findings of Spirale-skin impedance spectroscopy and the outcomes of the visual and auditory BCIs.

Fig. r2 | The impedance diagram of all channels for one subject.

Table r2. The impedance of all channels for all subjects in the visual test (kilohms).

	In-ear											Channels availability	Mastoid				Occipital						
	L 1	L 2	L 3	L 4	L 5	R 1	R 2	R 3	R 4	R 5	M1		M2	PZ	PO5	PO3	POZ	PO4	PO6	O1	OZ	O2	
S1	65.7	21.9	76.5	48.9	41.2	42.8	-	36.1	37.2	35.5	7/10	2.2	3.2	3.5	9.1	9.8	4.3	11.4	4.8	9.9	8.3	11.4	
S2	38.2	30.8	44.3	43.2	50.3	76.6	82.2	87.1	73.3	113.1	4/10	<1	1.4	10.3	4.3	2.6	11.9	11.5	6.5	8.1	10.4	9.8	
S3	44.0	33.5	27.4	36.3	15.7	40.4	39.5	110.0	42.7	25.1	9/10	5.2	<1	4.6	12.5	6.7	9.3	6.1	10.0	2.2	6.0	5.0	
S4	12.8	9.9	13.0	12.5	13.2	46.8	26.0	17.5	21.4	40.4	10/10	<1	4.9	1.2	7.2	7.5	3.5	<1	11.1	2.2	4.2	5.1	
S5	6.2	6.7	9.0	20.8	7.4	3.6	3.6	4.1	4.1	3.9	10/10	5.6	11.8	1.1	1.4	1.4	1.9	1.7	2.7	1.2	1.6	1.7	
S6	12.1	32.6	12.6	24.3	45.9	7.0	6.1	44.6	50.8	70.8	8/10	<1	4.9	3.4	6.0	2.2	3.3	4.5	7.3	3.2	6.4	2.9	
S7	15.7	20.6	7.6	9.5	49.7	29.1	11	15.2	9.4	65.2	9/10	<1	<1	9.5	21.8	21.8	14.7	20.4	20.6	22.0	13.1	26.7	
S8	11.9	5.6	8.6	7.7	<1	41.7	40.9	14.4	17.0	25.8	10/10	3.7	3.6	11.0	8.8	7.9	3.1	12.2	3.6	12.9	10.3	7.1	
S9	15.3	7.6	9.5	4.0	7.32	7.9	14.4	7.4	13.6	7.5	10/10	2.5	5.3	4.2	10.0	10.3	6.0	9	5.7	10.9	12.4	8.0	

I would like to reiterate our profound gratitude for your comments, and hope that the above explanation can effectively address the concerns and receive your approval.

Modification to the manuscript:

On lines 160-163, page 8, we have changed the sentence, “In addition, the impedance of SpiraleE at a sampling rate of 1 kHz is less than 1 kohm, which is comparable to the values of previously reported^{33,34}” to: “In addition, we observe that SpiraleE (where the electrode is a circle with a diameter of 1 mm) exhibits an impedance magnitude of less than 1 kilohms at 50 Hz. This finding is comparable to the results of previous reports^{33,34} and suggests that SpiraleE is suitable for use in EEG detection.”

On lines 238-240, page 12, we have added the sentence, “The impedance at 50 Hz has been determined to be 110.96 ± 23.31 kilohms across all in-ear channels, and the relevant data has been presented in Supplementary Table 1.”

References:

- 1 Bin, G., Gao, X., Yan, Z., Hong, B., & Gao, S. An online multi-channel SSVEP-based brain-computer interface using a canonical correlation analysis method. *J. Neural Eng.* **6**, 046002, (2009).
- 2 Chen, X., Wang, Y., Gao, S., Jung, T. P., & Gao, X. Filter bank canonical correlation analysis for implementing a high-speed SSVEP-based brain-computer interface. *J. Neural Eng.* **12**, 046008, (2015).

Comment 3

You make a reference to the electrodes reaching almost 50 degrees Celsius when placed inside the ear canal. While you say it is okay for short periods of time, it's unclear what 'short' means. S7 shows that the electrodes are indeed 49.3C inside the ear. The ear canal looks like it is close to 40C. It only takes a few minutes at that temperature to cause serious burns. Did your IRB allow you to do this?

Our response:

Thank you very much for your valuable comments. Regarding the comments on temperature, we agree that temperature is a crucial factor that needs to be carefully specified in our study. We

acknowledge that our previous description of the subject's sensations was solely based on a qualitative assessment, and we apologize for not providing a quantitative description. To address this issue, we have performed additional experiments and analyzed the data to provide a more precise and rigorous description of the temperature effects on participants.

First and foremost, we would like to reiterate that the well-being of our participants was our top priority throughout the study. We ensured that the proposed device was safe for use, and no participant suffered from any burn or injury during the experiment.

Acknowledging that external stimuli can cause SpiraleE to deform and raise its temperature over time, we offer the following clarifications:

(1) In order to quantitatively describe the duration of high-temperature exposure of SpiraleE in ears, we conducted *in vitro* experiments. We inserted a SpiraleE into a PE tube (which is transparent to infrared) with an inner diameter of 6.5 mm and applied electrothermal actuation using DC until the device was conformal to the inner wall of the tube. Throughout the experiment, we monitored the temperature of the entire field and SpiraleE's deformation using a Fluke Ti400 thermal imaging camera while simultaneously recording the time. The resulting time-temperature curve is presented in Fig. r3 (Supplementary Fig. 6), which also includes corresponding thermal images captured at various stages of the deformation process (marked with red borders). Specifically, the dotted line featured in the figure serves to indicate that despite the ongoing rise in temperature of the samples 2 and 3, the deformation process has completed in the preceding stage. These observations serve as validation of the high temperature duration required for the device to complete the deformation process. The original data derived from the thermal images captured at all points in the figure are included in the Source Data folder and have been jointly uploaded. Overall, the entire process lasted between 5 and 10 minutes, with the maximum temperature in the field exceeding 49 °C for approximately 30 seconds. These observations were found to comply with a number of established international standards, including those set forth by the International Electrotechnical Commission¹ and the European Committee for Electrotechnical Standardization², as explicitly demonstrated in Tables r3 and r4.

Fig. r3 | Time-temperature curves obtained during the deformation of SpiralE. Accompanying these curves are corresponding thermal images captured at various stages of the deformation process, as shown in the insets. Scale bars, 3 mm.

Table r3. IEC 60601-1: 2005 Table 23 and Table 24 – Allowable maximum temperatures.

Part		Contact duration for a time “t”	Metal and liquids	Glass, Porcelain, Vitreous Material	Plastic, Rubber, Wood, Molded Material
IEC 60601-1	Medical electrical equipment	$t < 1$ s	74 °C	80 °C	86 °C
		$1 \text{ s} \leq t < 10$ s	56 °C	66 °C	71 °C
		$10 \text{ s} \leq t < 1$ min	51 °C	56 °C	60 °C
		$1 \text{ min} \leq t$	48 °C	48 °C	48 °C
	Applied part having contact with patient	$t < 1$ min	51 °C	56 °C	60 °C
		$1 \text{ min} \leq t < 10$ min	48 °C	48 °C	48 °C
		$10 \text{ min} \leq t$	43 °C	43 °C	43 °C

Table r4. CENELEC Guide 29: 2007 Table A.1 – Burn threshold for longer contact times.

Material	Burn threshold T_s for			
	Contact period of	1 min	10 min	8 h and longer
		°C	°C	°C
Uncoated metal		51	48	43
Coated metal		51	48	43
Ceramics, glass and stone materials		56	48	43
Plastics		60	48	43
Wood		60	48	43

(2) Experimental investigations have been performed to quantify the temperature delivered by a high-temperature device placed on skin. In particular, a non-spiral device was affixed to the forearm, and a sheet-type thermocouple probe was positioned directly between the device and skin. The device was subjected to DC electrothermal actuation, which uniformly increased the temperature across the field to 50 °C, as determined *via* thermal imaging. Concurrently, the actual temperature of skin surface was measured using a thermocouple. Fig. r4a (Supplementary Fig. 7a) depicts the schematic diagram of the setup.

Analysis of the results revealed that when the maximum temperature of the field reached 50 °C (Fig. r4b, also known as Supplementary Fig. 7b), the temperature feedback from the thermocouple attached to the skin surface was ~40 °C (Fig. r4c, also known as Supplementary Fig. 7c). In conjunction with the brief duration of high temperature (≤ 30 s), this experience is no more intense than that of a fever for the participants, rendering it readily tolerable. This discrepancy was attributed to the presence of the SMP2 layer, EEG detection layer, and flowing air layer (as shown in Fig. 1c in the revised manuscript, exploded-view schematic illustration of the functional layers of the proposed SpiraleE) between the whole device and skin, as well as properties of the skin surface. Consequently, the temperature was significantly reduced upon reaching skin surface.

Additionally, the device was mounted on the skin for over 3 minutes and subsequently removed, revealing no redness or swelling at the attachment site (Fig. r4d, also known as Supplementary Fig. 7d). It should be noted, however, that the borders of the device secured with tapes may impede the airflow, thereby curtailing the rapid dissipation of heat and establishing restrictions of pressure. Nevertheless, such surface redness is easily and quickly restored. Our findings were further supported by subjective evaluations from 14 participants in visual and auditory BCI experiments, which demonstrated the safety of our device for use in ears.

Fig. r4 | Skin temperature evaluation during electrothermal test. a, Schematic diagram of the testing process. **b,** Thermal image of the device. **c,** Skin temperature test. **d,** Peeling off the device after 3 minutes. Scale bars, 10 mm.

(3) It is worth noting that due to differences in individual temperature perception, all subjects in our real tests were instructed to adjust the applied voltage autonomously to control the electrothermal deformation of the device. If any discomfort was experienced, the subjects were instructed to immediately turn the knob to cool the device. Upon lowering the temperature, we inquired of the subjects as to their willingness to continue with electrothermal actuation. In the event of disagreement, the experiment was promptly terminated. Conversely, if agreement was reached, the electrothermal actuation was resumed to drive ongoing deformation until Spirale conforms to the inner wall of the ear canal. Despite the extended duration of the experiment caused by repeated heating, we strived to maintain the safety of all subjects whilst ensuring optimal SNR. We monitored the entire deformation process using the Spirale-skin impedance to ensure that the device deformed sufficiently in ears without causing any discomfort. All participants in our study were able to successfully complete BCIs after undergoing single or multiple rounds of electrothermal actuation without reporting any discomfort.

Based on these observations, we are confident that our experiments are conducted safely and do not pose any risk of low-temperature burns.

Additionally, prior to the experiment, all participants were provided with a comprehensive description of the experimental protocol, and they signed the informed consent statement, which

included explicit language specifying that the experiment would be immediately discontinued in the event of discomfort or any other unforeseen circumstances. Notably, all participants completed the experiments without experiencing any discomfort or complications.

Furthermore, our experiment adhered to ethics (20220023) approved by the Institution Review Board of Tsinghua University, which governs non-invasive BCI researches. The study we conducted was carried out as part of the “Brain Computer Interface Contest” and was firmly within the scope of the ethics, thus ensuring its ethical suitability.

We hope that our responses above can address your concerns and apologize for any inconvenience.

Modification to the manuscript:

On lines 187-212, pages 10-11, we have added the sentences, “Subsequently, we proceed to insert a SpiralE into a polyethylene (PE) tube with an inner diameter of 6.5 mm, followed by electrothermal actuation until the device achieves conformity with the inner wall of the tube. This is performed to obtain a quantitative description of the high-temperature duration. Throughout the duration of the experiment, we employ a Fluke Ti400 thermal imaging camera to monitor both the temperature of the entire field and the deformation of SpiralE, while simultaneously recording time. The resultant time-temperature curves are presented in Supplementary Fig. 6, along with corresponding thermal images captured at various stages of the deformation process (indicated by red borders). Notably, the dotted line depicted in the figure indicates that although the temperature of samples 2 and 3 continue to rise, the deformation process has already been completed in the preceding stage. The experiments last between 5 to 10 minutes, with the maximum temperature in the field exceeding 49 °C for approximately 30 seconds, complying with a number of established international standards for allowable maximum temperatures on skin.

It should be noted that the actual temperature perceived by the skin is expected to be significantly lower than the maximum field temperature recorded by the thermal imaging camera, due to the presence of multiple layers including SMP2, EEGDL, and the convective air layer between the EEGDL and skin. In particular, a non-spiral device has been affixed to the forearm, and a sheet-type thermocouple probe is placed directly between the device and the skin. The device is subjected to electrothermal actuation, which uniformly elevates the temperature across the field to 50 °C, ascertained through thermal imaging (Fluke Ti400). Simultaneously, the temperature of

the skin surface is measured utilizing a thermocouple (Supplementary Fig. 7a). Upon reaching the peak temperature of 50 °C in the field (Supplementary Fig. 7b), the temperature feedback from the thermocouple affixed to the skin surface is found to be around 40 °C (Supplementary Fig. 7c), thereby confirming the safety of the device during contact with the skin. This data analysis suggests that the device can be utilized without any potential harm to the skin. Moreover, the device has been left attached to the skin for over 3 minutes at a maximum field temperature of 50 °C, and upon removal, no redness or swelling is observed at the attachment site (Supplementary Fig. 7d).”

On lines 231-234, page 12, we have changed the sentence, “The maximum temperature of Spirale in ear is 49.3 °C (Fluke Ti400), which is acceptable in a short period of time (Supplementary Fig. 7).” to: “In this study, we also employ a Fluke Ti400 thermal imaging camera to examine the maximum temperature of the device subsequent to insertion into the ear. Our results indicate that the maximum field temperature during Spirale’s deformation is 49.3 °C (Supplementary Fig. 10), while the temperature sensed by the skin is within the range of 40 °C, as confirmed by the above experiments.”

On lines 476-479, page 25, we have added the sentences, “Prior to the experiment, a comprehensive explanation of the experimental protocol was provided to all participants, and they signed the informed consent statement. As part of our study protocol, we afforded participants with the autonomy to withdraw from the experiment at any point in accordance with their own preferences, without incurring any untoward consequences.”

On lines 482-484, page 25, we have added the sentences, “In this process, participants were equipped with the ability to modulate the temperature elevation through manipulation of the voltage knob, thereby enabling them to terminate the experiment at their discretion.”

On lines 502-507, pages 26-27, we have changed the sentence, “All measurements were approved by the Medical Ethics Committee of Tsinghua University (20220023).” to: “The experimental techniques and procedures employed in this study were not expected to elicit any deleterious effects, whether physical, psychological, social, or otherwise, on the subjects. Owing to the non-invasive and well-tolerated nature of the experimental procedures, all participants were able to successfully complete BCIs without experiencing any discomfort or complications. All measurements were approved by the Institution Review Board of Tsinghua University (20220023).”

In addition, the numbering of figures has also been changed correspondingly.

References:

- 1 International Electrotechnical Commission (IEC). The 3rd edition of IEC 60601-1, (2005).
- 2 European Committee for Electrotechnical Standardization (CENELEC). Temperatures of hot surfaces likely to be touched // *CENELEC Guide 29*, (2007).

Reviewer #2:

Summary Comment

The authors have made sufficient revisions to improve the quality of this manuscript. This innovative work can meet the broad readership of *Nature Communications*, thus I would recommend the publication of this manuscript.

Our response:

We extend our heartfelt appreciation to the reviewer for these positive comments and recommendations on this work. The comments raised in the previous review have significantly enhanced the quality of our manuscript, and we are grateful for the contribution to our work. Once again, we express our gratitude to the reviewer for your time and efforts in providing us with such insightful suggestions.

Reviewer #3

Summary Comment

The authors have made a significant effort to address my comments. The proposed electrothermal activation and adjustability of shape and modulus in the ear are novel and robust. The authors have conducted a fair literature search to highlight the most significant advantages of SpiralE as an occlusion-free in-ear EEG device. I agree with the authors' response that devices like SpiralE can aid in finding in-ear channels with high SNR for future applications. The authors' definition of SNR (narrowband SNR) is rigorous and appropriate for SSVEP quality assessment. Overall, I believe that the revised manuscript presents exciting advancements and would be suitable for publication in *Nature Communications*. Here are some final suggestions:

Our response:

We are grateful to the reviewer for the positive feedback and recognition of our work. We have carefully considered the raised concerns and suggestions and made essential revisions to our manuscript. We sincerely hope that our responses meet the expectations and requirements of the reviewer.

Comment 1

In regards to the low modulus of 400 kPa and reliable resistance of electrodes under cyclic loading, it is evident that SpiralE is a mechanically reliable flexible electrode. However, it would be beneficial if the authors could provide a reference or calculation to support their claim that 400 kPa is soft enough for good conformability to the ear canal, for instance, by providing information on the representative curvature and Young's modulus of the human ear canal.

Our response:

We appreciate the valuable comments regarding SpiralE's compliance with the inner wall of the ear canal, which is a critical consideration when addressing this sensitive region. Accordingly, we utilized ABAQUS commercial software to assess the bending response of a simplified SpiralE during deformation in ear canal (Fig. r1a, also known as Supplementary Fig. 8a). With a designed inner radius of 3 mm (in the zero-stress state), SpiralE is compressed to 2.5 mm when constrained by the inner wall of the ear canal, assuming the radius of the ear canal is 2.5 mm. Notably, the interfacial stresses on the skin in contact with SpiralE (delineated by the dashed area) fall below the threshold for normal skin perception^{1,2} (20 kPa), as shown in Figs. r1b and c (Supplementary

Figs. 8b and c). Therefore, SpiraleE is unlikely to impose excessive loads on the inner wall of the ear canal during deformation.

In conjunction with the subjective feedback received from the 14 participants involved in the study, we have concluded that SpiraleE exhibits sufficient softness to achieve optimal conformability to the ear canal during the deformation process, while not imposing an unacceptable burden on such sensitive region.

We hope that the responses presented above can adequately address any questions or concerns regarding the conformability of SpiraleE and gain your valuable approval.

Fig. r1 | Summary of computational studies detailing the effects of normal and shear mechanical stresses from SpiraleE on human skin. **a**, Side profile view of SpiraleE attached to skin surface under bending deformation ($\alpha=\pi/3$). **b** and **c**, Finite element simulation results for a device applying normal and shear stresses on skin during bending deformations.

Modification to the manuscript:

On lines 213-216, page 11, we have added the following sentences, “Additionally, finite element analysis of a SpiraleE with an inner radius of 3 mm constrained by an ear canal with a radius of 2.5 mm shows that the interfacial stresses on the skin in contact with SpiraleE are within the normal skin perception threshold (20 kPa)^{35,36}, as depicted in Supplementary Fig. 8. As a result, SpiraleE is unlikely to impose an unacceptable burden on this sensitive region.”

In addition, the numbering of figures has also been changed correspondingly.

References:

- 1 Lee, C.H. *et al.* Soft core/shell packages for stretchable electronics. *Adv. Funct. Mater.* **25**, 3698-3704, (2015).
- 2 Lee, S.P. *et al.* Highly flexible, wearable, and disposable cardiac biosensors for remote and ambulatory monitoring. *npj Digital Med.* **1**, 2, (2018).

Comment 2

Regarding the skin contact impedance of 111 kohm, it seems acceptable when compared to recent ear EEG electrodes. Nonetheless, an impedance of less than 10 kohm at 50-60 Hz is recommended for standard EEG, especially for a passive electrode. To put this into perspective, several thin-film-based forehead EEG electrodes have achieved such low impedance with good electrode-skin conformability. This leads me to question whether full skin conformation in the ear canal is still unattainable with SpiraleE. It would be useful for the authors to address this limitation and provide insights for further improvement in the manuscript.

Our response:

We appreciate the reviewer's concern regarding the conformability and impedance value of our work. We have carefully considered this constructive feedback, as it offers additional opportunities to enhance our work.

In general, the impedance between the electrode and human skin is a critical factor that greatly impacts the quality of collected electrophysiological signals^{1,2}. The proposed SpiraleE exhibits higher impedance than prior studies that focused on thin film devices located on hairless areas³⁻⁵, such as the forehead and mastoid. This difference may be attributed to the following possibilities:

(1) The grease and stratum corneum on skin surface could affect the impedance value of the interface, thus skin cleaning is critical. While the hairless areas are easier to clean, the ear canal requires more effort due to its complex three-dimensional shape and semi-closed cavity. Although alcohol can moderately clean the inner wall of ear canal, the process is difficult and may lead to incomplete cleaning.

(2) In order to lower the impedance value, achieving close contact between the electrode and skin is crucial. While applying external force to parts such as the forehead, mastoid, and other open fields may easily ensure close contact of the interface, the contact between SpiraleE and ear canal relies solely on the stiffness of the device itself without additional assistance. Despite this

limitation, the achieved Spirale-skin impedance can successfully collect EEG signals to fulfill BCI applications across all subjects.

Therefore, we acknowledge that the current design has some limitations, and in the future, we can change the actuating method to actively intervene in the device's conformability to the inner wall of complex ear canal, reducing the interface impedance value.

Modification to the manuscript:

On lines 395-403, page 21, we have changed the following sentences, “In addition, signals detection and stimulus presentation are still based on commercial EEG recorder and LCD screen, which is not portable for day-to-day use outside the laboratory. Future work might also include efforts in building a flexible and wearable BCIs system by integrating Spirale with wireless modules and Augmented Reality glasses.” to: “Besides, the current design of Spirale places emphasis on a small contact area between the device and the inner wall of ear canal, as well as its hollow features, which may lead to an increased interface impedance due to the lack of external support to ensure contact force. In the future, it would be beneficial to balance the contact force guaranteed by external aids with the contact area and hollow characteristics by exploring alternative driving methods that can allow the device to actively adapt to the inner wall of ear canal. Moreover, the integration of Spirale with wireless modules and Augmented Reality glasses will eliminate the limitations of existing rigid and bulky devices and effectively facilitate the development of a flexible and wearable BCIs system with enhanced functionality and user experience.”

References:

- 1 Li, Z., Guo, W., Huang, Y., Zhu, K., Yi, H., & Wu, H. On-skin graphene electrodes for large area electrophysiological monitoring and human-machine interfaces. *Carbon* **164**, 164-170, (2020).
- 2 Wang, C. *et al.* On-skin paintable biogel for long-term high-fidelity electroencephalogram recording. *Sci. Adv.* **8**, eabo1396, (2022).
- 3 Norton, J. J. S. *et al.* Soft, curved electrode systems capable of integration on the auricle as a persistent brain-computer interface. *Proc. Natl. Acad. Sci. U. S. A.* **112**, 3920-3925, (2015).

- 4 Tian, L. *et al.* Large-area MRI-compatible epidermal electronic interfaces for prosthetic control and cognitive monitoring. *Nat. Biomed. Eng.* **3**, 194-205, (2019).
- 5 Shin, J. H. *et al.* Wearable EEG electronics for a Brain-AI Closed-Loop System to enhance autonomous machine decision-making. *npj Flex. Electron.* **6**, 32, (2022).

Comment 3

While I agree that the primary focus of the manuscript is to demonstrate Spirale's potential as a valid BCI electrode and not the machine learning model, it would be helpful if the authors could provide any differences between the channels or any significance in specific channels for decoding. Such findings could serve as a guideline for future in-ear BCI development.

Our response:

We appreciate the reviewer's concerns regarding the selection of channel number and location. Determining the optimal number and placement of sensors is a critical consideration in the field of BCI decoding. The importance of the electrodes is often depicted from the scope of spatial pattern, as the sensor correspond to the larger spatial weight are considered to play a significant role in decoding¹. The results have demonstrated that the temporal region, located near the ear, plays a crucial role in auditory attention decoding²⁻⁴. Therefore, in-ear sensors are highly appropriate for this paradigm.

The optimal number of electrodes required for effective BCI decoding is dependent on the chosen decoding methodology. In this study, we utilized a fundamental univariate algorithm and examined decoding accuracies as a function of an increasing number of channels. We observed that overall decoding accuracy generally improved with an increasing number of electrodes (Fig. r2). However, this univariate method is susceptible to unstable electrodes. Even with automatic data cleaning, unstable signals from certain electrodes can still impact the decoding results.

Consequently, while five channels per ear in this work may provide redundancy, the robustness of subjects and instability of some channels during the test may be addressed by interpolating with other channels for BCI applications. Due to the limited sample size of only five subjects, a definitive guideline for optimal electrode number could not be established. Nonetheless, based on our findings and experience, we recommend utilizing at least three electrodes per ear to construct an in-ear device. Furthermore, to fully exploit the information from multichannel recordings, we suggest the use of multivariate algorithms such as spatial filters³.

We regret that our study did not provide a definitive answer regarding the optimal number of channels required for in-ear BCIs. Nevertheless, our study represents an important reference for the application of BCIs based on in-ear EEG signals, and the selection of channels will be a critical focus of future research in the field of in-ear BCIs. We hope that our study will inspire further research into the optimal number of channels required for in-ear BCIs, and we are committed to contributing to this important area of study in our future work.

Fig. r2 | The relationship between the number of selected channels and the classification accuracy of auditory attention.

Modification to the manuscript:

On lines 392-395, page 21, we have changed the following sentence, “However, future investigations should consider a more in-depth examination of the neurophysiological significance of channels selection in ear, with the aim of achieving full coverage of in-ear EEG information while minimizing costs.” to: “Future research should conduct a more thorough investigation of the neurophysiological significance of channel selection in ears, to achieve comprehensive coverage of in-ear EEG information in various BCI paradigms while optimizing cost-effectiveness and ensuring robustness across subjects.”

References:

- 1 O'Sullivan, J. A. *et al.* Attentional selection in a cocktail party environment can be decoded from single-trial EEG. *Cereb. Cortex* **25**, 1697-1706, (2014).

- 2 Kulasingham, J. P., Joshi, N. H., Rezaeizadeh, M., & Simon, J. Z. Cortical processing of arithmetic and simple sentences in an auditory attention task. *J. Neurosci.* **41**, 8023-8039, (2021).
- 3 Geirnaert, S., Francart, T., & Bertrand, A. Fast EEG-based decoding of the directional focus of auditory attention using common spatial patterns. *IEEE Trans. Biomed. Eng.* **68**, 1557-1568, (2020).
- 4 Haufe, S. *et al.* On the interpretation of weight vectors of linear models in multivariate neuroimaging. *Neuroimage* **87**, 96-110, (2014).

REVIEWERS' COMMENTS

Reviewer #1 (Remarks to the Author):

Thank you for all of your hard work in answering my comments. You have provided an immense amount of patience, detail, and data. I have no further questions and believe the paper is ready for publication.

Reviewer #3 (Remarks to the Author):

The authors have effectively addressed the advantages, limitations, and future research directions in their revised manuscript. I think the manuscript is ready for publication now.

Responses to reviewers' comments

Reviewer #1

Summary Comment

Thank you for all of your hard work in answering my comments. You have provided an immense amount of patience, detail, and data. I have no further questions and believe the paper is ready for publication.

Our response:

We extend our gratitude for your valuable and constructive comments on this work, which have significantly enhanced the quality of our manuscript. Once again, we appreciate your time and efforts.

Reviewer #3

Summary Comment

The authors have effectively addressed the advantages, limitations, and future research directions in their revised manuscript. I think the manuscript is ready for publication now.

Our response:

We are grateful to the reviewer for the insightful comments and positive feedback on this work. Once again, we appreciate your time and efforts.